# Extending the Dynamic Wake Meandering Model in HAWC2Farm: A Comparison with Field Measurements at the Lillgrund Wind Farm

Jaime Liew[1], Tuhfe Göçmen[1], Alan W.H. Lio[1], and Gunner Chr. Larsen[1]

[1]Department of Wind Energy, Technical University of Denmark (DTU), Frederiksborgvej 399, 4000 Roskilde, Denmark

**Correspondence:** Jaime Liew (jyli@dtu.dk)

**Abstract.** With the increasing growth of wind farm installations, the impact of wake effects caused by wind turbines on power output, structural loads, and revenue has become more relevant than ever. Consequently, there is a need for precise simulation tools to facilitate efficient and cost-effective design and operation of wind farms. To address this need, we present HAWC2Farm, a dynamic and versatile aeroelastic wind farm simulation methodology that combines state-of-the-art engineer-
ing models to accurately capture the complex physical phenomena in wind farms. HAWC2Farm employs the aeroelastic wind turbine simulator, HAWC2, to model each individual turbine within the wind farm. It utilises a shared, large-scale turbulence box to represent atmospheric flow field effects at the farm level. The methodology incorporates a modified version of the Dynamic Wake Meandering model to accurately capture wake interactions. This approach not only ensures computational efficiency but also provides valuable insights for wind farm design and operation. To assess its performance, HAWC2Farm
is compared using time series extracted from field measurements at the Lillgrund wind farm, encompassing various scenarios involving wake steering via yaw control and a turbine shutdown. The results indicate that HAWC2Farm effectively addresses the challenges associated with modelling the complex dynamics within wind farms, thereby enabling more precise, informed, and cost-effective design and operation strategies.

## 1 Introduction

As the number and size of wind turbines in wind farm installations continue to grow, the impact of wake effects on power production, structural loads, and revenue remains a significant challenge to model. Accurate simulation tools are therefore in high demand to enable efficient and cost-effective wind farm and control design. Modelling the physical phenomena in a wind farm is complex and difficult due to the wide range of spatial and temporal flow scales involved. At each scale, nonlinear dynamics arise from a variety of factors, including microscopic material properties, fluid-structure interactions, and large-scale
atmospheric effects. Capturing all these phenomena accurately is impractical and computationally expensive. In this study, we present the HAWC2Farm aeroelastic wind farm simulation platform, which combines state-of-the-art engineering models into a dynamic and versatile tool for simulating wind farm performance. Individual wind turbine structures, atmospheric flow effects at the farm level, turbine and farm controllers, and wake interactions are all modelled in a computationally efficient way, providing valuable insights for wind farm design and operational applications.

Several categories of wind farm simulation environments exist, each with different objectives, levels of detail, and computational costs. Steady-state wind farm simulators make up the majority of available tools, with applications in wind farm layout and control optimisation for power maximisation (Riva et al., 2020). Such tools, which can execute a single wind farm simulation in the order of nanoseconds, include PyWake (Pedersen et al., 2019) and FLORIS (NREL, 2021) with several state-of-the-art computationally low-cost wake models. Steady-state models are unable to resolve dynamic flow interactions between turbines, which is important in the design of closed-loop wind farm control strategies. For these tasks, quasi-dynamic wind farm simulators, such as FLORIDyn (Becker et al., 2022), LongSim (Bossanyi et al., 2022), SimWindFarm (Grunnet et al., 2010), WFSim (Boersma et al., 2018), and OnWaRDS (Lejeune et al., 2022b, a), use low-fidelity rotors and wake profiles in a time-marching simulation. Such tools are suitable for simulating closed-loop control strategies, but due to the simple rotor model, are unable to resolve mechanical loading effects on the turbine structure without additional modelling. High-fidelity Computational Fluid Dynamics (CFD) simulations, such as Ellipsys3D (Hodgson et al., 2021; Sørensen et al., 2015) and SOWFA (Fleming et al., 2014) can resolve the flow-field evolution through a wind farm at a high level of detail. Furthermore, by coupling a CFD solver with aeroelastic wind turbine models, such as in the vortex solver, MIRAS (Ramos-García et al., 2021), turbine operational characteristics and structural dynamics can be determined. Given the high computational demands of CFD simulations, simulating large wind farms or running numerous simulations to optimise wind farm layout and control can become impractical. For this reason, the genre of medium fidelity *aeroelastic wind farm simulations* comes into focus. Aeroelastic wind farm simulators use aeroelastic wind turbine models in a simplified turbulent flow field compared to CFD. By doing so, the level of detail in the wind farm flow is relinquished in exchange for reduced computational requirements. Available aeroelastic wind farm simulators include FAST.Farm (Jonkman et al., 2018), and the currently presented, HAWC2Farm (Liew et al., 2022).

HAWC2Farm couples the aeroelastic turbine simulator, HAWC2 (Larsen and Hansen, 2007; Madsen et al., 2020), with a modified interpretation of the Dynamic Wake Meandering (DWM) model (Larsen et al., 2008), which is capable of scaling to wind farm simulations consisting of hundreds of turbines. A synthetic turbulence box is propagated through the wind farm, typically using the Mann turbulence model (Mann, 1994, 1998). All components are implemented in a time-marching manner at a high temporal resolution, typically between 10Hz and 100Hz. This opens doors to many use cases, such as quantifying the structural response (*e.g. resonance or fatigue*) of each turbine under non-stationary or transient wake effects. Furthermore, advanced control strategies can be implemented in a realistic dynamic setting.

The simulation methodology is qualitatively compared against measurement data collected from the Lillgrund offshore wind farm (Sood et al., 2022). Collected SCADA and LIDAR data are used to design HAWC2Farm simulations to recreate two scenarios in the Lillgrund wind farm. The first scenario takes place over an eight-hour period with a non-stationary wind direction, during which a yaw misalignment test was conducted on the wind farm. This scenario is of interest as the periodic changes in yaw angle can be detected in downstream turbines due to wake deflection. The second scenario is a four-hour period, in which the turbines equipped with load sensors are aligned. Additionally, one of the upstream turbines shuts down during this period, allowing for a sudden step change in turbine thrust to be recreated in HAWC2Farm and compared to the field

measurements. The presented comparison extends and consolidates the verification against Large Eddy Simulations performed by Liew et al. (2022).

In this study, we present the HAWC2Farm aeroelastic wind farm simulation methodology, which is described in detail in Section 2, with a focus on the implementation of the DWM model. In particular, novel changes to the wake meandering and wake profile solvers are outlined. The field measurements from the Lillgrund wind farm used in this study are described in Section 4, along with the corresponding simulation setup in HAWC2Farm. The results from the HAWC2Farm simulations are then compared to the Lillgrund measurements in Section 5, and the paper concludes with final remarks and recommendations for future work.

## 2 Methodology

This section describes the underlying models used in HAWC2Farm to perform aeroelastic wind farm simulations. HAWC2Farm unifies three components: the wind turbine, the turbulent wind field, and the wakes. Each of these components is a dynamic model, able to march forward in time.

### 2.1 Aeroelastic turbines

The aeroelastic turbines in this study are simulated using parallel instances of HAWC2 (Larsen and Hansen, 2007; Madsen et al., 2020), with each instance representing a single turbine in the wind farm. HAWC2 is a multi-body finite element code with an aerodynamic front-end written in FORTRAN and has been modified to expose several functions to Python using C-compliant interfaces (Horcas et al., 2020). Before each time step, controller set points and high-resolution wind field data are passed to HAWC2, and it returns an instantaneous axial induction profile and turbine sensor data to the wake components and wind farm controller, respectively. The HAWC2 turbine model can include a turbine controller that interprets set points provided by the wind farm controller, if in use. HAWC2 provides high-resolution time-series simulations of the turbine, including operating conditions (*i.e.* power output, rotor speed, and blade pitch angles) and structural loads.

### 2.2 The collective wind field

The collective wind field in this simulation synthesises all aspects of the flow within and around wind farms, including ambient atmospheric boundary layer turbulence, wind shear, wind direction changes, wake deficits, and wake-induced turbulence. It is updated after both the HAWC2 and DWM layers. A large, high-resolution turbulence box is pre-generated and incrementally advected at each time step. Accurately calculating turbine fatigue loads requires using a turbulence box cell size smaller than 0.02 times the turbine diameter, D, in all spatial directions as recommended by Liew and Larsen (2022). The Mann turbulence model is also recommended, as it effectively incorporates fundamental turbulence physics with limited input demands while remaining computationally and memory efficient Mann (1998). Alternatively, high-fidelity precursor fields from Large Eddy Simulations (LES) can also be used, such as in Liew et al. (2022). While the frozen turbulence box is typically propagated at

90 a constant speed and direction, HAWC2Farm allows for modification of both the speed and direction of the turbulence box propagation. Wind direction changes in simulations often require careful consideration of fluid conservation laws (Stieren et al., 2021), but when performed gradually, a simple rotation of the turbulence box can provide valuable insight into the effects of non-stationary inflow on the wind farm system.

## 2.3 Dynamic Wake Meandering model

95 The Dynamic Wake Meandering (DWM) model is a crucial component of the HAWC2Farm simulation platform. The DWM model unifies three typical characteristics of a turbine wake in its model as illustrated in Fig. 1: the wake meandering, the wake profile, and added wake turbulence. To simulate the large-scale motion of the wake, a series of passive wake tracer particles are employed, which meander through the turbulent wind field. As these particles advect, the wake profile (depicted in blue) evolves based on the distance travelled. Additionally, the model tracks the wake-induced turbulence weighting factor profile

100 (depicted in red), which represents the extent of additional turbulence introduced by the wake-producing rotor.

The definition of the DWM model in the IEC 61400 international standards (International Electrotechnical Commission, 2005) allows for flexibility in its implementation, as it does not specify details such as the numerical method for solving the wake profile or the method of filtering low frequencies in the passive tracer motion. Additionally, in contrast to the standard defini-

105 tion, the extended formulation of the DWM model presented here explicitly incorporates wake deflection. In this study, several modifications and extensions to the DWM model are proposed to accommodate the aeroelastic turbines and collective wind farm flow field, while still respecting the original definition (Larsen et al., 2008).

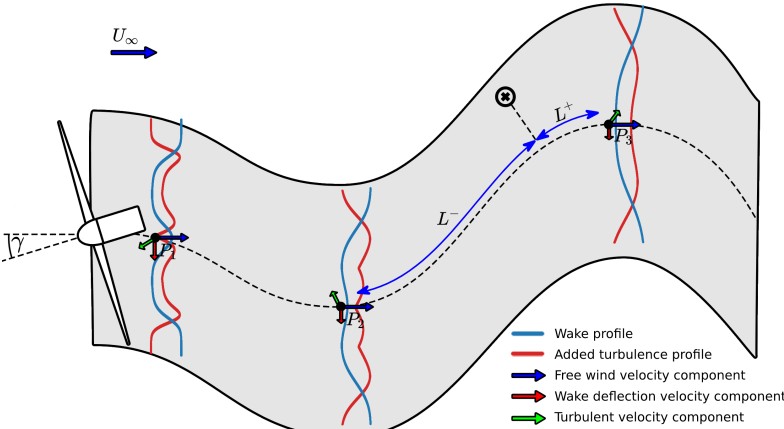

**Figure 1.** Illustration of the various components in the DWM model, including the tracer particles ($P_1$, $P_2$, and $P_3$), the wake profile (blue), the added wake turbulence profile (red), and an example of a velocity interpolation at a point in space, x.

### 2.3.1 Deficit profile solver

The axisymmetric thin shear layer approximation of the Navier-Stokes equations can be expressed as two partial differential equations representing momentum and mass conservation, respectively:

$$U\frac{\partial U}{\partial x} + V_r\frac{\partial U}{\partial r} = \frac{1}{r}\frac{\partial}{\partial r}\left(\nu_T r\frac{\partial U}{\partial r}\right) \tag{1}$$

$$\frac{1}{r}\frac{\partial}{\partial r}(rV_r) + \frac{\partial U}{\partial x} = 0 \tag{2}$$

where $U$ and $V_r$ are shorthand for $U(x,r)$ and $V_r(x,r)$, representing the longitudinal and radial velocities at radial distance $r$ and downstream distance $x$ respectively, and $\nu_T$ is the eddy viscosity, which varies with $x$ depending on the chosen definition of the wake model (Reinwardt et al., 2018). A Neumann boundary condition is found at $r=0$ to replicate a reflection, and a fixed Dirichlet boundary condition as $r \to \infty$ indicates that the flow converges to the free wind speed far from the wake centre. Additionally, the boundary condition at rotor position $x=0$, $U_0(r)$, is determined on the axial induction profile of the rotor at a given moment as:

$$\left.\frac{\partial U(x,r)}{\partial r}\right|_{r=0} = 0 \tag{3} \qquad \lim_{r\to\infty} U(x,r) = 1 \tag{4} \qquad U(0,r) = U_0(r) \tag{5}$$

where it is assumed that $U$ and $V_r$ are normalised by the free wind speed. The wake profile can be solved numerically in a step-wise manner. The numerical methods for solving the DWM deficit profile vary in literature in terms of discretisation and computational efficiency. Most finite difference schemes described in DWM literature use an explicit solver (Keck et al., 2015, 2012; Madsen et al., 2010). Madsen uses a 5-point stencil with forward differencing (Madsen et al., 2010). Keck, instead, uses three-point central differencing in the radial direction and forward differencing in the $x$ direction (Keck et al., 2015, 2012). These methods can face numerical instabilities due to the nature of the forward Euler method. In this section, both a backward and forward Euler method for solving Eq. 1 is outlined, as well as a justification for using the implicit solver method based on numerical stability requirements.

First, Eq. (1) and (2) can equivalently be expressed in the following more convenient forms:

$$U\frac{\partial U}{\partial x} = \left(\frac{\partial \nu_T}{\partial r} + \frac{\nu_T}{r} - V_r\right)\frac{\partial U}{\partial r} + \nu_T\frac{\partial^2 U}{\partial r^2} \tag{6}$$

$$rV_r = -\int_0^\infty r\frac{\partial U}{\partial x}dr \tag{7}$$

**Table 1.** Partial derivative substitutions for explicit and implicit Euler schemes along the $x$ axis with central differencing along the $r$ axis.

| Variable | Explicit Euler | Implicit Euler |
|---|---|---|
| $\dfrac{\partial U}{\partial x}$ | $\dfrac{U_{i+1,j}-U_{i,j}}{\Delta x}$ | $\dfrac{U_{i+1,j}-U_{i,j}}{\Delta x}$ |
| $U\dfrac{\partial U}{\partial x}$ | $U_{i,j}\dfrac{U_{i+1,j}-U_{i,j}}{\Delta x}$ | $U_{i,j}\dfrac{U_{i+1,j}-U_{i,j}}{\Delta x}$ |
| $\dfrac{\partial U}{\partial r}$ | $\dfrac{U_{i,j+1}-U_{i,j-1}}{2\Delta r}$ | $\dfrac{U_{i+1,j+1}-U_{i+1,j-1}}{2\Delta r}$ |
| $\dfrac{\partial^2 U}{\partial r^2}$ | $\dfrac{U_{i,j-1}-2U_{i,j}+U_{i,j+1}}{\Delta r^2}$ | $\dfrac{U_{i+1,j-1}-2U_{i+1,j}+U_{i+1,j+1}}{\Delta r^2}$ |

Next, by discretising along the $x$ and $r$ axes by the respective step sizes, $\Delta x$ and $\Delta r$, the discrete notation for the velocities is $U_{i,j} = U(i\Delta x, j\Delta r)$ and $V_{r,i,j} = V_r(i\Delta x, j\Delta r)$. Using the derivative substitutions in Table 1, the explicit formulation for Eq. (6) is

$$U_{i+1,j} = U_{i,j} + \frac{\Delta x}{U_{i,j}}\Big((-C_1 - C_2)U_{i,j+1} + 2C_1 U_{i,j} + (C_2 - C_1)U_{i,j-1}\Big), \tag{8}$$

where

$$C_1 = -\frac{\nu_T}{\Delta r^2} \tag{9}$$

$$C_2 = \frac{1}{2\Delta r}\left(V_{r,i,j} - \frac{\nu_T}{r}\right). \tag{10}$$

Eq. (8) can be explicitly solved given the previous wake state at step $i$. Similarly, the implicit scheme is formulated as follows:

$$\underbrace{(C_1 - C_2)}_{a_j}U_{i+1,j-1} + \underbrace{\left(\frac{U_{i,j}}{\Delta x} - 2C_1\right)}_{b_j}U_{i+1,j} + \underbrace{(C_1 + C_2)}_{c_j}U_{i+1,j+1} = \underbrace{\frac{U_{i,j}^2}{\Delta x}}_{d_j} \tag{11}$$

The given linear system is represented as a tridiagonal system in Eq. (12), where the coefficients $a_j$, $b_j$, $c_j$, and $d_j$ are used in Eq. (11). To solve this tridiagonal system, a specialised tridiagonal solver algorithm can be employed. One such solver routine is LAPACK's `xgtsv` function, which is specifically designed to efficiently handle systems of linear equations with symmetric positive definite tridiagonal matrices. This function offers several advantages over general-purpose solver routines, including

efficient memory usage and reliable numerical stability (Anderson et al., 1999).

$$
\begin{bmatrix}
b_0 & c_0 & 0 & \dots & & 0 \\
a_0 & b_1 & c_1 & \ddots & & 0 \\
0 & a_1 & \ddots & \ddots & & \vdots \\
\vdots & \ddots & \ddots & \ddots & & c_{N-2} \\
0 & 0 & \dots & a_{N-2} & & b_{N-1}
\end{bmatrix}
\begin{bmatrix}
U_{i+1,0} \\
U_{i+1,1} \\
\vdots \\
U_{i+1,N-2} \\
U_{i+1,N-1}
\end{bmatrix}
=
\begin{bmatrix}
d_0 \\
d_1 \\
\vdots \\
d_{N-2} \\
d_{N-1}
\end{bmatrix}.
\tag{12}
$$

The boundary condition far from the centre (Eq. (4)) is enforced by setting the system coefficients $a_{N-2} = 0$, $b_{N-1} = 1/\Delta x$, and $d_{N-1} = U_{i,N-1}/\Delta x$. Similarly, the root boundary condition (Eq. (3)) is met by setting $c_0 = 2C_1$. Both implicit and explicit schemes solve for $V_r$ in Eq. (7) by iteratively integrating Eq. (2) from the centre outwards using trapezoidal rule integration:

$$
r_{j+1}V_{r,i,j+1} = r_j V_{r,i,j} - \frac{\Delta r}{2\Delta x}\left(r_j(U_{i+1,j} - U_{i,j}) + r_{j+1}(U_{i+1,j+1} - U_{i,j+1})\right)
\tag{13}
$$

where $V_{r,i,0} = 0$. A radial boundary of 3R, where R denotes the rotor radius, was found adequately large to accommodate the width of the wake in most scenarios but may need to be increased depending on turbulence conditions and the size of the wind field domain.

While both explicit and implicit methods are capable of solving the DWM deficit equation, the implicit scheme is numerically stable for a wider range of discretisations of $\Delta r$ and $\Delta x$. Ensuring numerical stability is crucial in the presented application due to the presence of noise in the axial induction profile boundary condition obtained from the aeroelastic turbine simulation. The turbulence and transient effects in the boundary condition can introduce fluctuations, which, if not properly handled, may result in unstable solutions for the wake deficit. This risk of numerical instability is especially pronounced in long and turbulent simulations, where noisy axial induction profiles are more likely to trigger an instability, highlighting the importance of maintaining numerical stability throughout the analysis. The stability was empirically tested over a range of longitudinal and radial discretisations. At each discretisation, 50 random axial induction profiles were introduced as boundary conditions to the deficit flow solver using both the explicit and implicit scheme to identify if an instability was triggered (Fig. 2). The random profiles consisted of random axial induction values along the rotor ranging from -1 to 1. The explicit solver presented a narrow stability region, whereas the implicit solver was numerically stable when $\Delta x \gtrsim 25\Delta r^2$ and $\Delta r < 1$. Although the implicit solver takes approximately twice as long to perform an iteration on the DWM wake profile, the explicit solver is only stable for radial discretisations of less than 8 points per radius, making the explicit solver unsuitable to represent the rotor induction at arbitrary resolutions. Furthermore, the additional computational time is negligible in comparison to the full wind farm simulation. The extra computational cost in solving the tridiagonal system was, therefore, seen as a necessary compromise to ensure numerically stable wake profiles. For this reason, HAWC2Farm is built on the implicit wake deficit solver.

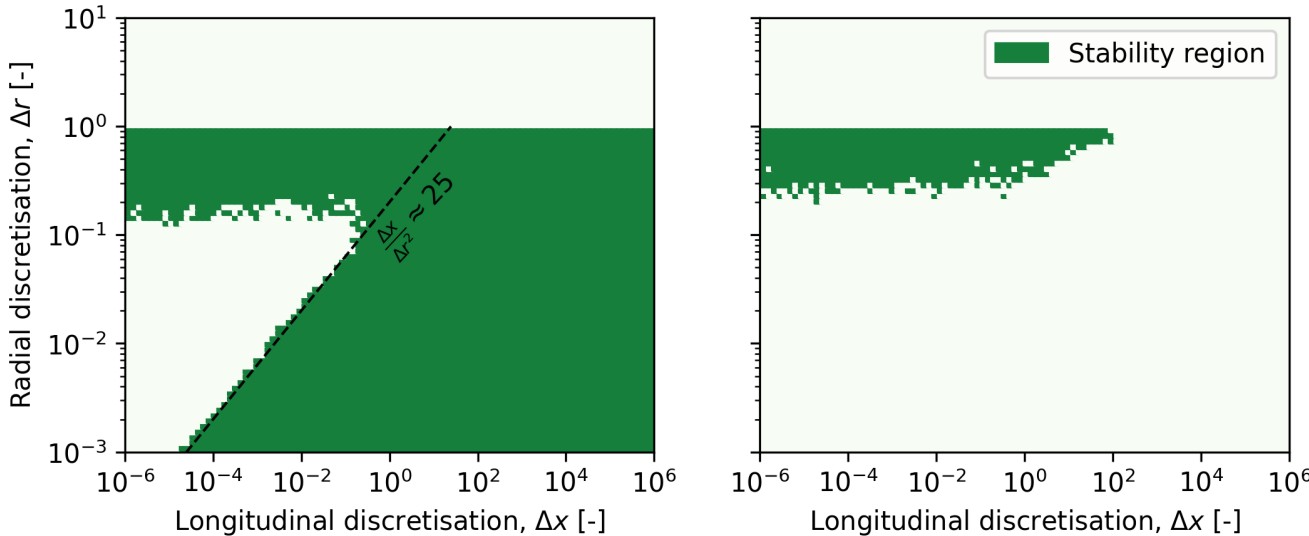

**Figure 2.** Approximate stability regions of the implicit (left) and explicit (right) solver schemes.

### 2.3.2 Wake-induced turbulence

In addition to the wake profile is a corresponding small-scale turbulence field defined by a wake-induced turbulence weighting factor profile, $k_{mt}$. As formulated in Madsen et al. (2010), $k_{mt}$ is determined by the depth and the shear of the wake deficit, taking the form of:

$$175 \quad k_{mt}(x,r) = |1 - U(x,r)|\, k_{m1} + \left|\frac{\partial U(x,r)}{\partial r}\right| k_{m2} \tag{14}$$

where $k_{m1}$ and $k_{m2}$ are tunable parameters. Eq. (14) can be readily evaluated from the longitudinal wake deficit and its derivative. To apply the added weight turbulence to a wind field, a highly resolved unit variance isotropic turbulence field is superimposed over the ambient wind field with a weighting equal to $k_{mt}$. Being linked to the wake deficit, the wake-induced small-scale turbulence field is meandered along with the wake deficit. This is identical to the methods described by Madsen et al. (2010) and Larsen and Hansen (2007). An example of $k_{mt}$ is shown in Fig. 3 (right).

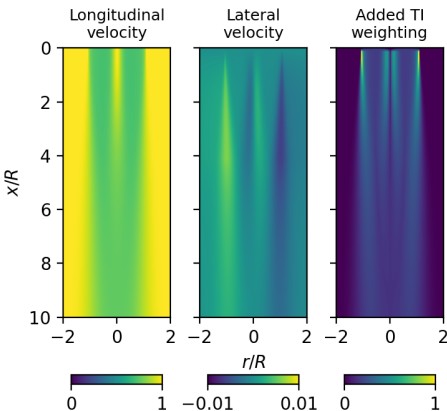

**Figure 3.** Slices of wakes generated using the implicit wake profile solver. The flow propagates from the top of the figure (i.e. the rotor plane) to the bottom. Slices of the longitudinal (left) and lateral (middle) velocities are shown as well as the wake-induced turbulence weighting factor profile (right).

### 2.3.3 Meandering with filtering

The large-scale meandering of the wake deficit is modelled by warping its path as it advects through the turbulent wind field. The DWM model uses a concept described as *passive tracers* by Larsen et al. (2008), and more recently, *observation points* by Lejeune et al. (2022b), Gebraad and Van Wingerden (2014), and Becker et al. (2022). Passive tracers are emitted from the turbine rotor, endowed with turbine axial induction and orientation information. They advect transversely and vertically according to the large spatial scales of the turbulent wind. At each time step, the wake profile described in Section 2.3.1 is solved based on the incremental downstream distance that the 'wake particle' has travelled, $\Delta x$. Larsen et al. (2008) defines the wake meandering velocity to be a spatial average of the wind field velocity over either the rotor disk or more correctly, the instantaneous expanded wake deficit area.

In past implementations of HAWC2, this low pass filtering was attempted by using a low-resolution turbulence box, where the grid spacing was equal to 1D (Larsen and Hansen, 2007; Madsen et al., 2010). By doing so, the Nyquist frequency of the turbulence box would be equal to the intended cut-off frequency. This, in combination with a linear interpolator on the turbulent wind field, provides a crude approximation of the desired low pass filter with a slow roll-off (Fig. 4, green). A more comprehensive approach to conducting the filtering involves utilising a spatial filter, where wind speeds are evenly sampled across a disk perpendicular to the direction of the free stream flow. However, this method can be computationally demanding, as each particle iteration may necessitate hundreds or even thousands of wind field samples to carry out the spatial filtering.

In the presented methodology, a temporal filter is used in place of the spatial filter. The cutoff frequency of the temporal filter is set to approximately $f_c = U/(16D)$ to match the cutoff frequency of the spatial filter (Fig. 4, orange). This is, as expected, somewhat lower than the upper cut-off frequency limit introduced in Larsen et al. (2008) (i.e., $f_c = U/(2D)$), and more in line

with full-scale field observations reported in Lio et al. (2021). To determine the value of $f_c$ for the temporal filter, a spatial filtering procedure is employed on the turbulence box. This involves uniformly sampling points across the rotor disk area at different longitudinal distances within the box. By analysing the frequency response of the spatial filter, it becomes possible to identify the 3dB cut-off frequency, which is widely employed in the field of signal processing as a reference point denoting a 50% decrease in signal amplitude. This cut-off frequency is subsequently utilised as the value for the temporal filter. The cutoff frequency may differ from case to case depending on the dimensions and properties of the turbulence box used.

The advantage of the temporal filter is that it only requires a single sample of the wind field per time step. Compared to the spatial filter, which requires orders of magnitude more samples per time step, the temporal filter can save computational effort while giving a comparable frequency response to the original definition.

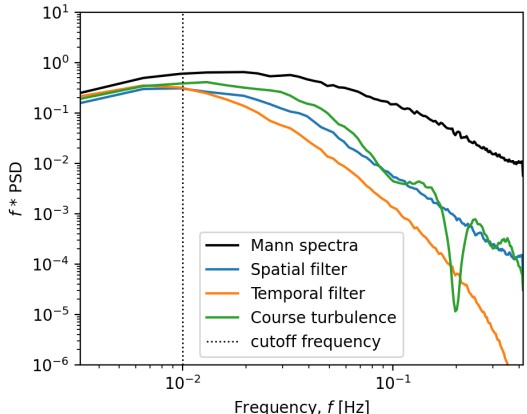

**Figure 4.** Frequency response of the longitudinal turbulent wind speed from Mann-generated turbulence subject to different filtering techniques.

The temporal filter, illustrated in orange in Fig. 4, is achieved with a first-order infinite impulse response digital low pass filter applied to the turbulent wind field using the recursive equation:

$$\overline{u}_k = (1-\alpha)\overline{u}_{k-1} + \alpha u_k \tag{15}$$

$$\overline{v}_k = (1-\alpha)\overline{v}_{k-1} + \alpha v_k \tag{16}$$

$$\overline{w}_k = (1-\alpha)\overline{w}_{k-1} + \alpha w_k \tag{17}$$

where $\overline{u}_k$, $\overline{v}_k$, and $\overline{w}_k$ are, respectively, the filtered longitudinal, lateral, and vertical wind speeds measured at the location of the passive tracer at time step $k$. The discrete filter coefficient, $\alpha$ is a constant related to the desired cutoff frequency, $f_c$, and time step, $\Delta t$:

$$\alpha = \cos(2\pi f_c \Delta t) - 1 + \sqrt{\cos^2(2\pi f_c \Delta t) - 4\cos(2\pi f_c \Delta t) + 3} \tag{18}$$

The passive tracer location can then be updated in 3D space using the recursive relation:

$$x_k = x_{k-1} + \Delta t \overline{u}_k \tag{19}$$

$$y_k = y_{k-1} + \Delta t (\overline{v}_k + v_{\text{deflect},k}) \tag{20}$$

$$z_k = z_{k-1} + \Delta t \overline{w}_k \tag{21}$$

where $v_{\text{deflect}}$ is the lateral wake velocity due to wake steering as described in the next section.

### 2.3.4 Meandering with wake deflection

Given the axisymmetric nature of the wake, Hill's vortex theory can be used to estimate the deflection of a wake tracer particle when a turbine is misaligned with the free wind (Branlard, 2020). As proposed by Larsen et al. (2020), a Hill's vortex analogy of the wake induction field can be incorporated to estimate wake deflection as follows:

$$v_{\text{deflect},k} = \frac{dy_k}{dt} = -0.4 U_{def,k} \sin \gamma \tag{22}$$

where $v_{\text{deflect},k}$ is the lateral tracer velocity at time $k$, used in Eq. (20), $\gamma$ is the yaw misalignment of the rotor at the moment that the passive tracer is emitted, and $U_{def,k}$ is the rotor-average wake deficit of the wake tracer, which can be determined on the axisymmetric wind field as:

$$U_{def,k} = 1 - \frac{2}{R^2} \int_0^R r U_k(r) dr \tag{23}$$

### 2.4 Wake summation

If multiple wakes overlap, a point-wise summation is performed to determine the wind velocity at a point in space. The recommended superposition method for the DWM model varies in the literature. Common methods include *dominant* wake summation (Eq. (24)), in which only the strongest wake is considered:

$$U(x,y,z) = \min_i (U_i(x,y,z)) \tag{24}$$

where $U_i(x,y,z)$ is the single wake wind speed of turbine $i$ at position $(x,y,z)$, and $U(x,y,z)$ is the aggregated wind speed. Additionally, there are linear summation and quadratic summation described by Eq. (25) with $k=1$ for linear and $k=2$ for quadratic.

$$\left(1 - \frac{U(x,y,z)}{U_\infty}\right)^k = \sum_{i=1}^n \left(1 - \frac{U_i(x,y,z)}{U_\infty}\right)^k \tag{25}$$

where $U_\infty$ is the ambient wind speed, and $n$ is the number of wakes. It should be noted that these summation methods do not require knowledge of the number of upstream turbines as the summation is performed point-wise. The IEC 61400 international standards recommend using the dominant deficit below rated wind speed, and linear summation above rated wind

speed (International Electrotechnical Commission, 2005). Larsen et al. (2013) used successfully the dominant wake method in a field validation study based on measurements from the Dutch offshore wind farm Egmond Ann Zee. Later, Larsen et al. in 2015 introduced the linear summation approach based on a full-scale load study on the Lillgrund offshore wind farm with a focus on high inflow wind speeds (Larsen et al., 2015).

The dominant wake deficit is further validated for the DWM model by Reinwardt (2022), in which a comprehensive vali-
dation with field measurements on the Curslack wind farm in Germany is performed. In this study, both quadratic and linear summation methods were found to overestimate the wake deficit, with the linear summation occasionally producing negative wind speeds, particularly in scenarios with several overlapping wakes. Based on the outcomes of these studies, the dominant wake summation method is used for the remainder of the presented analysis. However, it should be noted that all mentioned summation methods are implemented in the HAWC2Farm platform.

## 3   Code overview

The HAWC2Farm aeroelastic wind farm simulator is designed to accurately model the dynamic turbine interactions within a wind farm. It combines the use of HAWC2 aeroelastic turbine models with the DWM model to simulate the response of the turbines and the wind farm flow field. The HAWC2 model provides a detailed representation of the geometry, aerodynamics, control, and structural dynamics of the turbines, allowing for a more accurate simulation of their behaviour. The DWM model
is used to simulate the propagation of the turbine wakes, providing a dynamic boundary condition for the simulation.

The code is parallelised using MPI, allowing for efficient and accurate simulation of the complex interactions between the turbines and the wind as shown in the simulation loop flow diagram in Fig. 5. In the code, each turbine-wake pair is executed in parallel, and 3D segments of the collective wind field with wakes are periodically communicated to each turbine. Each turbine model internally propagates the wind field segment until a new wind field segment is provided. The interval at which
updates to the wind field occur is determined by the turbulence level in the field. A range of 1 to 5 seconds has been found to be suitable, striking a balance between reducing the overhead of inter-communication and avoiding sudden discontinuities in the wind field. To maintain simulation stability, the code implements various measures, including infrequent stepping of the wakes, to satisfy the implicit stability condition outlined in Section 2.3.1. This section highlights that taking larger steps in the longitudinal direction helps circumvent areas prone to numerical instability.
Parallel execution of turbine and wake calculations significantly accelerates the code, but the remaining performance bottle-neck arises from intercommunications between processes. The computational time of HAWC2Farm, running at a simulation frequency of 100Hz on the DTU Sophia HPC cluster (Technical University of Denmark, 2019), is depicted in Figure 6. For a small number of wind turbines, the simulation takes approximately 2 to 3 times longer than real-time, which aligns with individual HAWC2 simulation durations on this HPC system. However, as more turbines are included, the ratio of elapsed
real-time to simulation-time increases linearly, with an approximate rate of 0.06s/s per additional turbine. Due to the presence of 32 cores per node in the HPC system, simulations that involve more than 32 turbines necessitate inter-node communication.

However, this additional communication does not pose a problem, as the computational time maintains a linear scaling even with 128 turbines. This highlights the noteworthy scalability of HAWC2Farm for large wind farm simulations.

Overall, the HAWC2Farm aeroelastic wind farm simulator is a powerful and user-friendly tool for analysing the performance of wind farms. It provides a detailed and accurate representation of the dynamic turbine interactions within a wind farm, enabling a better understanding of the factors that impact the performance of wind farms in terms of both production and structural loading under different control settings.

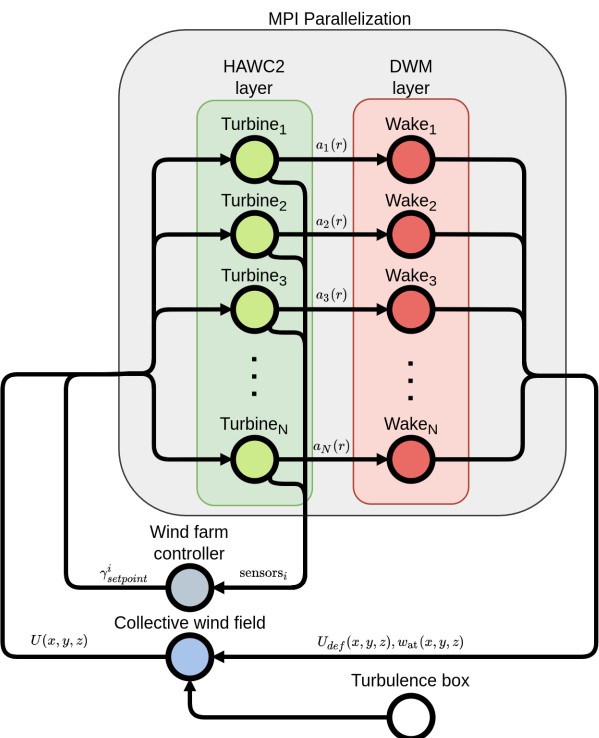

**Figure 5.** Flow diagram of HAWC2Farm iteration structure with parallelisation. $a_i(r)$ is the axial induction profile as a function of rotor radius, $r$, of the $i$th turbine. $\gamma_i$ is the controller set point (*e.g.* yaw or induction) for the $i$th turbine. $U_{def}$ and $w_{at}$ are the the wake deficit profile and wake-induced turbulence weighting factor profile, respectively.

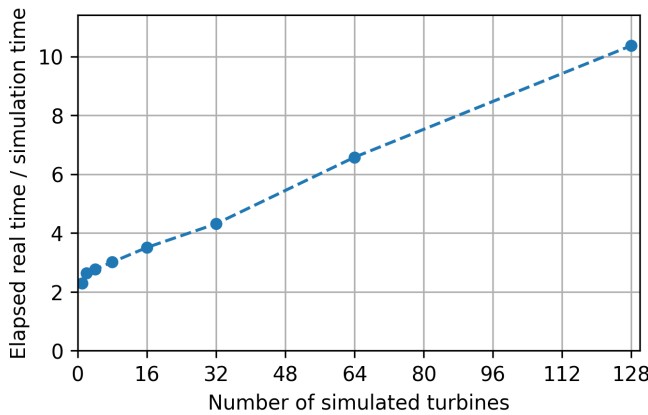

**Figure 6.** Elapsed real time to simulation time ratio of HAWC2Farm simulations with varying number of simulated turbines. (sampling rate: 100Hz. simulation time: 10 minutes. HPC: DTU Sophia HPC cluster (Technical University of Denmark, 2019).)

## 4  Lillgrund measurements and simulation setup

To demonstrate the utility of the HAWC2Farm for wind farm modelling, several scenarios measured in the Lillgrund offshore
wind farm are recreated in simulation. The Lillgrund offshore wind farm consists of 48 bottom-fixed 2.3MW turbines located
in Øresund between Denmark and Sweden. The turbine inter-spacing ranges from 3.3D to 4.3D.

The measurement campaign was conducted as part of the EU TotalControl project to validate high-fidelity codes (Sood
et al., 2022). The campaign was running during the period from September 2019 to February 2020, and included flow field
measurements using LIDARs as well as turbine SCADA data and structural loads based on strain-gauge measurements. As for
the flow field observations, three long-range pulsed scanning Doppler wind LIDARs (Vasiljević et al., 2016), were installed on
the Lillgrund wind turbine transition pieces. Only one of the LIDARs is used in this study, facing upstream of turbine B08 (see
Fig. 7) and recording the flow field at various altitudes within the wind farm. To measure blade deformation, strain gauges are
located 1.5m away from the blade roots. Additionally, two strain gauges are installed on the tower, both located at a height of
8.52m from the tower base. These strain gauges are placed 90 degrees apart from each other around the circumference of the
tower, enabling the measurement of the tower bending in two different directions.

**Table 2.** Summary of turbine information including the Case scenario it is used for, availability of structural load measurements and SCADA data, and normalisation factor used in Eq. 26 and Eq. 27.

| Turbine | Cases | Yaw sequence | Load sensors | SCADA data | Normaliser for: |
|---------|-------|:---:|:---:|:---:|:---:|
| B06 | 1, 2, 3 | ● | ● | ● | Case 2 |
| A05 | 2 | | | ● | |
| D07 | 1 | | ● | ● | Case 1 |
| B07 | 3 | | ● | ● | |
| B08 | 3 | | ● | ● | Case 3 |

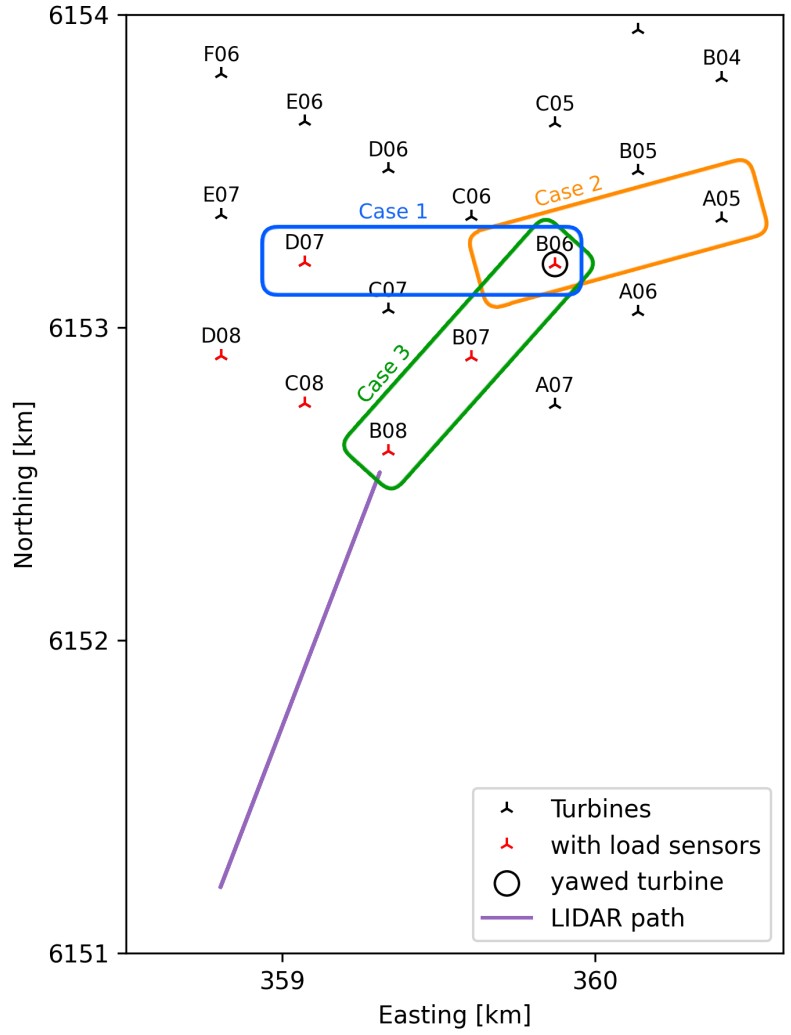

**Figure 7.** A subsection of the Lillgrund wind farm layout with measurement turbines and LIDAR scanning paths indicated.

Based on data availability between the LIDARs, SCADA data and structural load measurements, two scenarios are recreated in HAWC2Farm. While all 48 turbines are simulated in both scenarios, selections of turbines, which are analysed and presented in this study, are divided into three distinct cases as illustrated in Fig. 7. Scenario 1 includes Cases 1 and 2, which analyse different turbines over the same eight-hour time period when the yaw misalignment campaign takes place. Case 1 investigates two turbines, which are equipped with load sensors, whereas Case 2 investigates the most downstream turbines of a row of four, which transitions between partial and full wake cases as the incoming wind direction changes. Scenario 2 includes Case 3, which covers a four-hour period where the wind is aligned with the selected turbines. Case 3 is unique as all analysed turbines are equipped with load sensors, and the middle turbine undergoes a shutdown event, allowing the transient wake step to be investigated at both the shutdown and downstream turbines. Turbine information and measurement availability are summarised in Table 2.

To recreate the two scenarios, the LIDAR and SCADA data are used to design the wind conditions of the wind farm. A long-range pulsed scanning Doppler wind LIDAR mounted at the base of turbine B08 scans along a line facing away from the wind farm as illustrated in Fig. 7, and is used to determine the inflow conditions (wind direction, wind speed, wind shear, and turbulence intensity) to be simulated. The LIDAR measurements have a sampling time of 30 seconds and are collected over 72 points ranging from 14m to 212m in height. The simulation setup is also assisted by SCADA data of the turbines of interest with a sampling time of 2 seconds. For example, Scenario 1 draws from the nacelle direction measurements of turbine B06 to determine the yaw sequence during the yaw misalignment campaign, and the turbine status signal is used in Scenario 2 to determine the time of the turbine shutdown event.

## 5   Results

In this section, two scenarios, as described in Section 4, are recreated using HAWC2Farm and the resulting time series are compared with the field observations. For each scenario, a short description of the simulation is provided, followed by an analysis of the time series comparison.

The results presented in this section are normalised to account for differences in the measurement offset and scaling calibration. The power output and wind speed time series measurements use a relative normalisation with the mean value of an upstream turbine:

$$\hat{x}_i = x_i - \bar{x}_0 \tag{26}$$

where $x_i$ is the absolute quantity for turbine $i$, $\hat{x}_i$ is the normalised quantity for turbine $i$, and $\bar{x}_0$ is the mean value observed or simulated at the most upstream turbine. These normalising upstream turbines differ between the three cases and are defined in Table 2. The only quantities that use a different normalisation are the structural load measurements, which are additionally normalised by the standard deviation of the signal as:

$$\hat{x}_i = \frac{x_i - \bar{x}_0}{\sigma_0} \tag{27}$$

where $\sigma_0$ is the standard deviation of the most upstream turbine. The motivation behind employing a distinct normalisation technique for the structural loads is to account for possible calibration errors in the strain gauge measurements and potential discrepancies in the definition of structural properties within the HAWC2 model for the 2.3MW turbines at Lillgrund. This normalisation procedure ensures that the resulting signal has a zero mean and unit variance, facilitating meaningful comparisons even in scenarios involving span error and zero shift in the strain gauge calibration. As a result, the load measurements become independent of the calibrated zero value and the modulus of elasticity of the strain sensors, enabling a more accurate analysis of the load trends. It is important to acknowledge that these normalisations are intended to facilitate qualitative comparisons between measured and simulated time series. A further validation study would be required to accurately assess potential systematic errors.

To distinguish between these two normalisation methods, the units in Fig. 10, 11, and 14 are prepended with a $\Delta$ (*e.g.* [$\Delta$ m/s], [$\Delta$MW]) when using the relative normalisation (Eq. 26), and as [-] when using the scaled normalisation (Eq. 27).

## 5.1 Lillgrund wake steering campaign with wind direction change

In the following section, the results of recreating Scenario 1 are presented. The purpose of Scenario 1 is to examine the effects of a yaw misalignment sequence and wind direction changes on turbine performance. To do this, an eight-hour period is simulated, in which the yaw misalignment of turbine B06 is varied while the wind changes direction. We will use the HAWC2Farm simulation tool to recreate these conditions based on measurements from the Lillgrund wind farm and compare the results to the measurements from two sets of turbines: Case 1 and Case 2. The operational conditions and loads of the turbines will be analysed to determine any differences.

### 5.1.1 Simulation set up

In this simulation, the wind farm is modelled using LIDAR measurements to determine the inflow wind field. The wind direction changes from westerly to south-westerly over an eight-hour period, with an approximately linear rate of change as shown in Fig. 8. The wind speed remains relatively constant at around 11.0m/s at hub height, with a power-law shear exponent of 0.135 at a reference height of 65m. Turbulence is generated using the Mann model with a grid spacing of 0.02D, as recommended in previous research to ensure unbiased load calculations (Liew and Larsen, 2022). The turbulence intensity level is specified by setting $\alpha\epsilon^{2/3} = 0.01$ (where $\alpha \approx 1.7$ is the spectral Kolmogorov constant, and $\epsilon$ is the rate of viscous dissipation of turbulent kinetic energy) based on measured LIDAR time series (TI$\approx 9\%$), while other parameters are set based on IEC standard values (i.e., eddy lifetime parameter $\Gamma = 3.9$ and length scale L = 33.6m). The turbulence box dimensions and discretisation are $(L_x, L_y, L_z) = (322336, 3000, 115.93)$m and $(N_x, N_y, N_z) = (262144, 2048, 64)$ respectively, and the simulation is run for a duration of 8 hours at a sampling rate of 100Hz.

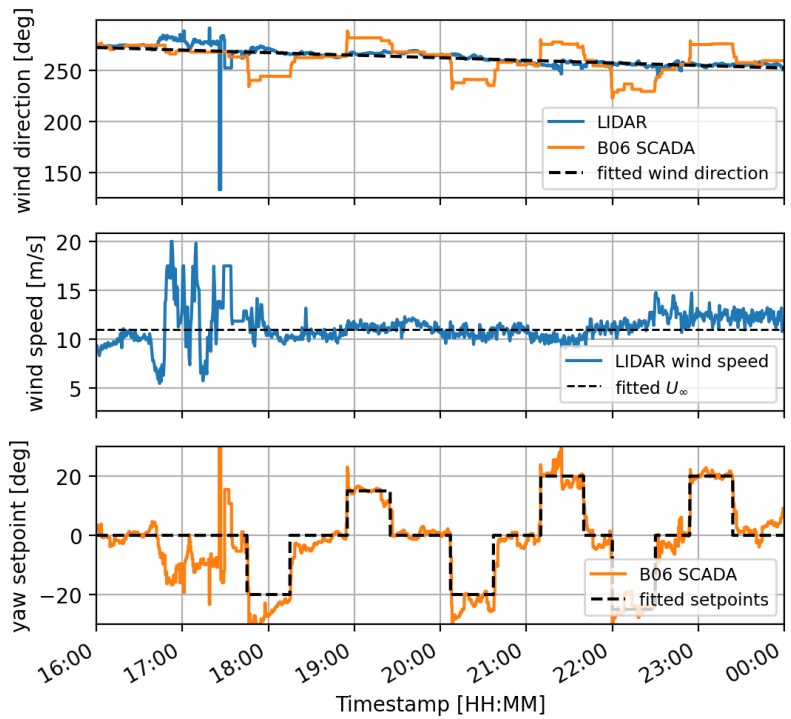

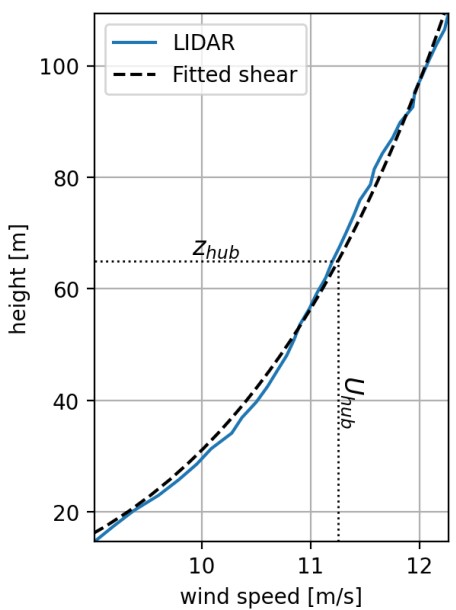

**Figure 9.** Measured wind shear profile from LI-DAR for Scenario 1 (Case 1 and 2).

**Figure 8.** Input LIDAR and SCADA data for Scenario 1 (Case 1 and 2).

The yaw sequence for turbine B06 during this period is determined using operational log data and SCADA signals, as shown in Table 3. This sequence is visible in the SCADA data of the turbines, shown in Fig. 8. The max yaw rate and yaw acceleration of turbine B06 are set to $0.2\mathrm{deg/s}$ and $0.1\mathrm{deg/s}^2$.

**Table 3.** Yaw sequence executed by turbine B06 in Case 1 and Case 2.

| Step start time [HH:MM:SS] | Step duration [minutes] | Step amplitude [deg] |
|---|---|---|
| 17:45:00 | 30 | -20 |
| 18:55:00 | 30 | 15 |
| 20:07:00 | 30 | -20 |
| 21:10:00 | 30 | 20 |
| 22:00:00 | 30 | -25 |
| 22:54:00 | 30 | 20 |

### 5.1.2 HAWC2Farm comparison with measurements

First, we consider Case 1, which focuses on turbines D07 and B06 as defined in Fig. 7. In this field scenario and simulation, the incoming wind is initially from the west, causing a full wake interaction between turbines D07 and B06. This can be seen in the time series outputs of both the field measurements and the HAWC2Farm simulation in Fig. 10 at 16:00, where the power output of B06, located downstream, is notably lower compared to D07. As the wind direction shifts counter-clockwise, the two turbines exit the wake scenario, causing the power output of B06 to match that of D07. Although the measurements indicate

this occurs earlier than in the simulation, this discrepancy is likely due to the linear modelling of the wind direction change as well as unmodelled fluctuations in the wind speed at approximately 17:00. An alternative explanation could be that there are variations in the propagation speed of the wakes. However, considering the significant duration that has elapsed during the initial stages of the case and the relatively small distance between the two turbines, it is improbable that this factor contributes significantly to the observed phenomena.


The yaw misalignment sequence for B06 starts at 17:45 causing a power reduction at the controlled turbine and is shown to match the measurements and simulation. The tower side-side moment also corresponds to the yaw misalignment in both the measurements and simulation, indicating good agreement. During the misalignment periods, the measured wind speed of B06 drops, which is expected as the measurement is taken on the nacelle, which is turned away from the incoming wind. In

contrast, the HAWC2Farm wind speed does not show a decrease as the wind speed measurement takes place at a fixed location and orientation.

The most significant yaw step, the fifth, takes place at 22:00, causing a clear drop in the power of B06 in the HAWC2Farm outputs. However, it is unclear if this power drop is present in the measurements, as the power output of B06 is rapidly increas-

ing over this period due to the difference in local wind direction at the end of the scenario (22:00 to 00:00). The HAWC2Farm simulation continues to rotate the wind field, causing B06 to enter the wake of D08 and C07. However, as the wind direction change appears to cease in the measurements and the wind speed slightly increases, this causes the power output of B06 to temporarily increase, making the effect of the yaw step less clear in the SCADA measurements.

Overall, Fig. 10 shows good agreement between HAWC2Farm results and the field measurements of wind direction, wind

speed, and structural loads at the controlled turbine B06 for Case 1. However, there are notable discrepancies for turbine D07, where the simulation shows the turbine operating at rated power throughout the measurement period, while the measurements indicate occasional reductions in power output. This likely arises from differences in how the wind speed and direction vary between the simulation environment and in the field, which might relate to large-scale turbulent structures not currently included in the calculations (Alcayaga et al., 2022).

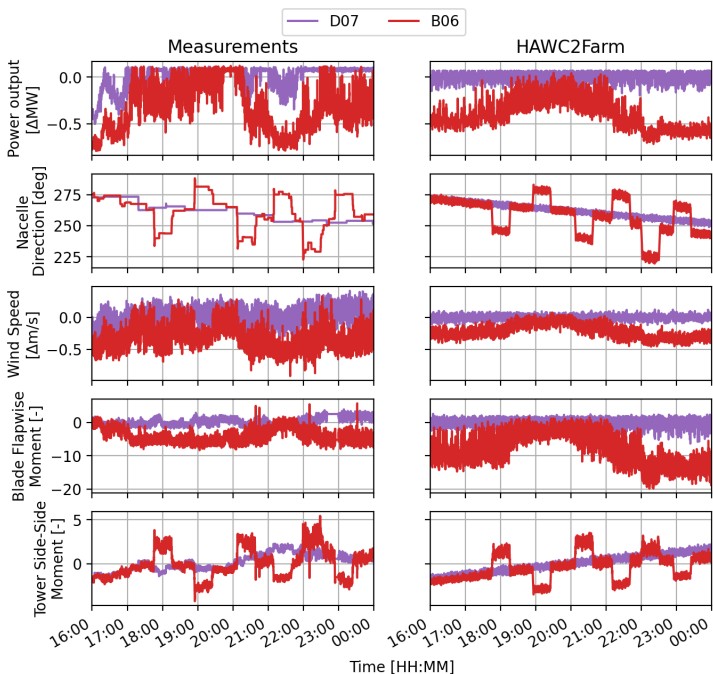

**Figure 10.** Case 1 time series results from both SCADA measurements and the HAWC2Farm simulation.

In Case 2, which takes place during the same time frame as Case 1, the focus is on turbines B06 and A05. As the wind direction changes, turbine A05 is affected by the wakes from several rows of turbines, leading to significant fluctuations in its power output in the first half of the period, as shown in Fig. 11. During the second half of the period, turbine A05 is in the wake of turbine B06, which is also undergoing its yaw sequence. The HAWC2Farm simulation in Fig. 11 shows that during the fifth yaw step at 22:00, B06 experiences a drop in power, while A05 experiences an increase in power. This illustrates the effects of wake steering. This effect can also be seen in the wind speed signal at the same moment. However, the field measurements do not show the effects of wake steering, as both turbines are close to their rated power and there is a difference in wind direction and a slightly higher wind speed in the field. Unfortunately, load measurements for A05 are not available, so it is not possible to gain further insight into the impact of upstream yaw control on its performance.

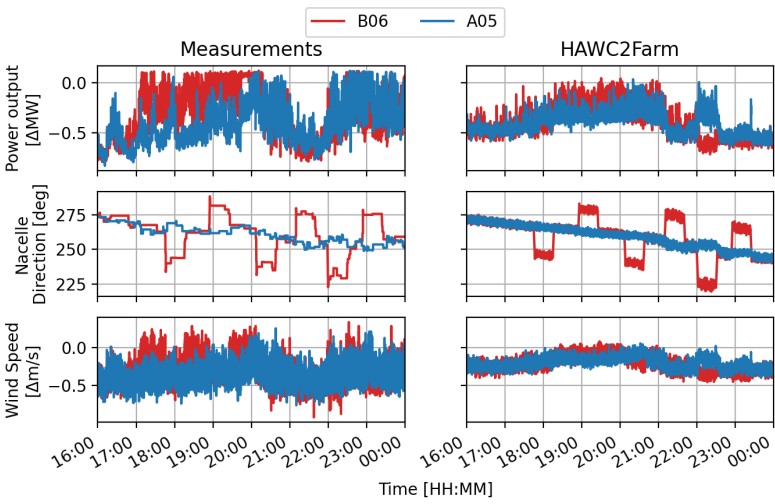

**Figure 11.** Case 2 time series results from both SCADA measurements and the HAWC2Farm simulation.

## 5.2 Lillgrund full wake scenario with turbine shutdown

The following section presents the results of Scenario 2, which is a four hours simulation with relatively constant wind direction and speed. A unique feature of this period is the sudden shutdown of turbine B07, which is recreated in the corresponding HAWC2Farm scenario. The simulated time series for selected turbines (Case 3) during this sudden step are compared to measurement data.

### 5.2.1 Simulation set up

In Case 3, the wind parameters are determined using LIDAR measurements in a similar way to Case 1 and Case 2. The wind direction and hub speed remain relatively constant over the target period, as shown in Fig. 12, and are set to $222^o$ and 10m/s, respectively. The power-law shear exponent is fitted at 0.105 (Fig. 13). The turbulence intensity level is specified by $\alpha\epsilon^{2/3} = 0.02$, based on measured LIDAR time series (TI$\approx 9\%$), while other parameters are set based on IEC standard values (i.e. eddy lifetime parameter $\Gamma$ = 3.9 and length scale L = 33.6m). The turbulence box dimensions and discretisation are
$(L_x, L_y, L_z) = (145548, 3000, 115.93)$m and $(N_x, N_y, N_z) = (262144, 2048, 64)$ respectively, and the simulation is run for a duration of 4 hours at a sampling rate of 100Hz.

By analysing the SCADA data from turbine B07, the exact moment at which the turbine shuts down can be determined at 15:52:30. This timestamp is used in the HAWC2Farm simulation to accurately model the shutdown of the simulated B07 turbine.

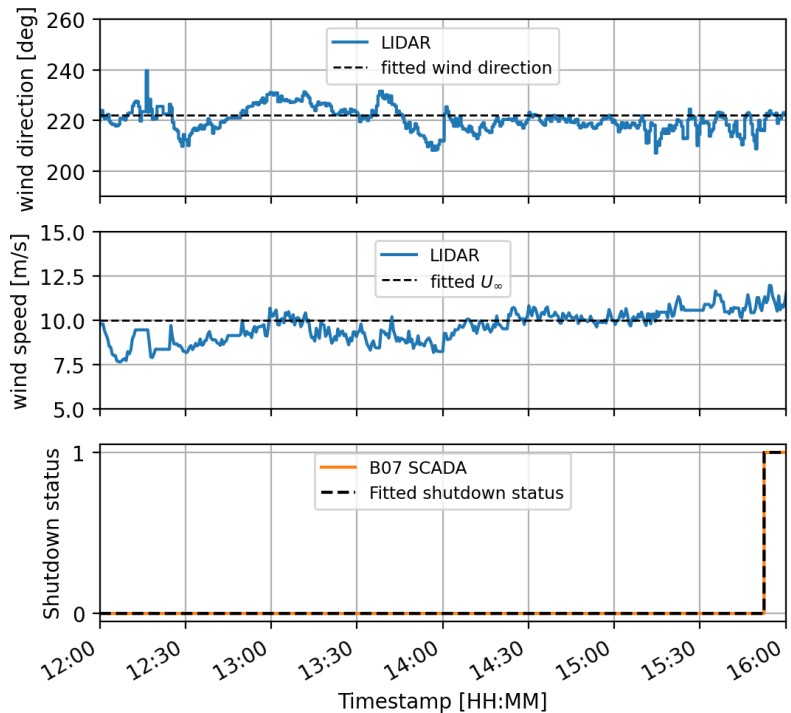

**Figure 12.** Input LIDAR and SCADA data for Case 3.

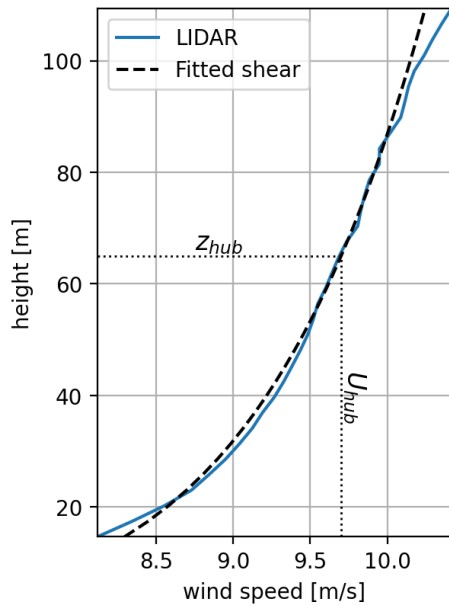

**Figure 13.** Measured wind shear profile from LI-DAR for Case 3.

### 5.2.2 HAWC2Farm comparison with measurements

In Case 3 (Fig. 14), the wind flows parallel to turbine row B, creating a full-wake situation. During this period, the second turbine in the row, B07, experiences a sudden shutdown, causing a step change in its power, blade flapwise moment, and tower moment measurements. This step is apparent in both measurements and the HAWC2Farm simulation, with similar roll-off periods and step direction, though there may be some variation in step magnitude.

Turbine B06 is located downstream of B07 making it susceptible to the influence of B07's wake when B07 experiences a sudden change in performance. In the HAWC2Farm simulation, the power, wind speed, and blade flapwise moment of B06 also show a sudden step in the same direction and at the same time as in the measurement time series, indicating that the wake propagation from B07 matches the changes in B06. However, the tower side-side moment at B06 does not display a distinct step change, indicating that the alterations in B07 do not impact the side-side moment of B06 in a comparable manner.

The magnitudes of the steps in the power, wind speed, blade flapwise moment, and tower moment at the downstream turbines B07 and B06 vary between HAWC2Farm simulations and the field measurements. Similar to Case 1 and Case 2 results, such differences are mainly driven by a potential lack of detail in turbine representation (including its response under several operating conditions), turbulence modelling and resolution of wake effects. However, for a medium fidelity wind farm simulator, HAWC2Farm is demonstrated to be highly capable of reproducing and capturing trends in the most relevant

quantities of interest (particularly for wind farm flow control), as well as transient dynamics both in terms of applied controlled settings and associated response of the turbines in large wind farms under non-stationary flow and multiple wake effects.

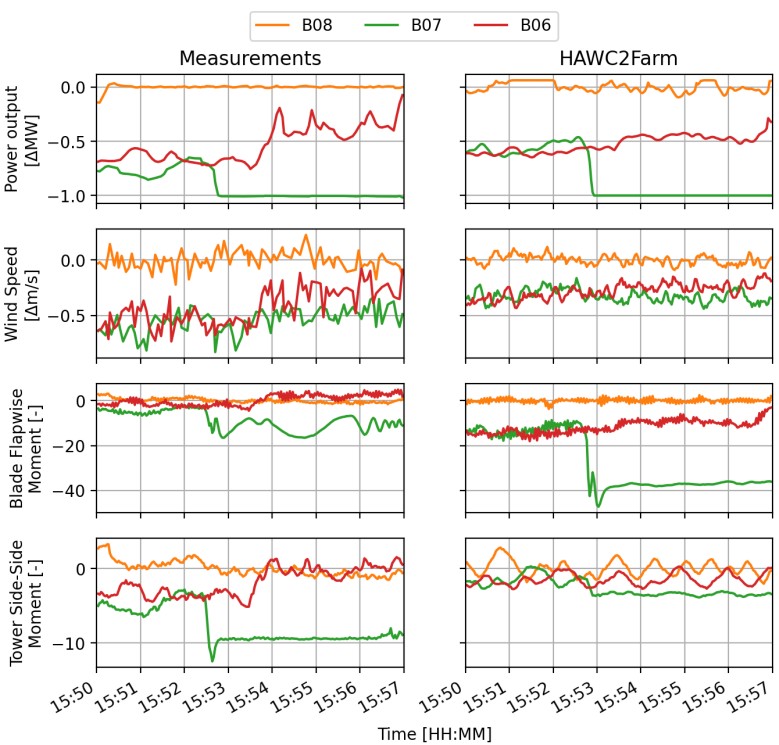

**Figure 14.** Case 3 time series results from both SCADA measurements and the HAWC2Farm simulation.

## 6 Conclusions and recommendations

The HAWC2Farm aeroelastic wind farm simulation methodology provides a versatile approach for modelling the complex and dynamic physical phenomena in wind farms. By combining state-of-the-art engineering models, HAWC2Farm can accurately
capture the performance of individual wind turbines, the collective wind field, and wake interactions within a wind farm even when these are subjected to complicated transient flow phenomena. The method is computationally efficient, enabling the simulation of large wind farms with aeroelastic information from all turbines. Specific details are provided on how the Dynamic Wake Meandering (DWM) model is implemented to accommodate large wind farm simulations. These details include a numerically stable formulation of the wake profile solver, a real-time wake meandering filtering, which can advect through arbitrary
resolutions of the background turbulence, and an implementation of the yaw-dictated wake deflection model. The results of the HAWC2Farm simulation are compared against selected full-scale measurements from the Lillgrund wind farm, showing good agreement for turbine operating conditions, loads, and wake effects. This was achieved by setting up two HAWC2Farm simulations with durations of 8 and 4 hours at a simulation sampling rate of 100Hz.

Nevertheless, there are several potential areas of improvement in both the modelling methodology and the creation of the measurement data. Firstly, the DWM method assumes an axisymmetric wake. Recent studies show that a curled wake shape can manifest from turbines with yaw misalignment Martínez-Tossas et al. (2019). The impact on the downstream turbines in terms of performance and loads may be better represented by modifying the wake profile to reflect these non-axisymmetric effects. Furthermore, the concept of a wake centre location breaks down when multiple wakes overlap. While it is computationally convenient to merge the strings of passive tracers using a wake summation model, the method is highly challenging to validate from measurements as no coherent wake centre can be measured. With adequate tuning of the DWM parameters, the wake effects may be sufficiently represented for structural load or power estimation, but the true nature of the wake is only an approximation. Further verification of the method is needed using higher fidelity flow simulations and detailed LIDAR measurements.

Recreating scenarios observed in the field in a simulation setup is not a trivial task, especially when it comes to predicting the behaviour of complex non-stationary flows. In the particular scenarios presented, one of the main challenges is to verify the wake deflection of a yawed downstream turbine while the incoming wind direction changes. This is because the complex interactions between the turbines and interchanging conditions between full and partial wakes make it difficult to identify trends and accurately model the system. In addition, differences between the turbine models used in the simulation and the real-world turbines can lead to discrepancies between the simulated and actual response of the wind farm. The lack of load sensors downstream of the yawed turbine makes it difficult to determine how the loads on the turbine are influenced by an upstream yaw-controlled turbine. This uncertainty can impact the accuracy of the simulation comparison. Therefore, further comparisons with full-scale measurements are required to validate and calibrate the tool with regard to structural loading calculations. Nevertheless, HAWC2Farm shows comparable time series results in the presented comparison, particularly in the dynamic propagation of wakes and turbine operational conditions during changing turbine and wind conditions. Case 3, consisting of a turbine shutdown, was recreated successfully in HAWC2Farm, showing the correct timing between the turbine shutdown and the delayed effects on the downstream turbine. The blade flapwise and tower side-side moments in both the shutdown and downstream turbines showed matching trends, with some discrepancies in the magnitudes which can be attributed to ambiguities in the calibration of the load sensors.

Overall, the presented aeroelastic wind farm simulation methodology, HAWC2Farm, is shown to have great potential in testing and evaluating wind farm flow control strategies such as de-rating and wake steering under dynamic and non-stationary conditions, providing insight into power production as well as the structural load implications of spatial and dynamic variations of the wind field, and simulating complex scenarios such as turbine shutdown events or wind direction changes.

*Code and data availability.* The HAWC2Farm source code is available open-source at doi.org/10.5281/zenodo.8028485 (Liew, 2023a), and the underlying DWM implementation is available at doi.org/10.5281/zenodo.8028555 (Liew, 2023b).

*Author contributions.* JL developed and implemented the HAWC2Farm software, and performed the analysis of results. TG defined and processed the measurement data used in the investigation. GCL and JL developed the theory behind the extended DWM model. JL, TG, AWHL and GCL contributed to the conceptualisation, investigation, and reporting of the research presented in this paper.

*Competing interests.* The authors declare that they have no conflict of interest.

*Acknowledgements.* The authors would like to thank Anders Sommer from Vattenfall A/S for the original provision of the Lillgrund SCADA data and Elliot Simon from DTU Wind Energy for initial post-processing and management of the data-base. This research has been supported by the European Commission, Horizon 2020 Framework Programme (TotalControl (grant no. 727680)). Additionally, the authors gratefully acknowledge the computational and data resources provided on the Sophia HPC Cluster at the Technical University of Denmark, DOI: 10.57940/FAFC-6M81Technical University of Denmark (2019).

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
