# Peer review of "Extending the Dynamic Wake Meandering Model in HAWC2Farm: A Comparison with Field Measurements at the Lillgrund Wind Farm"

_Wind Energy Science, 2023_

## Referee Comment (RC1)

**Review of "Extending the Dynamic Wake Meandering Model in HAWC2Farm: Lillgrund Wind Farm Case Study and Validation", manuscript id: wes-2023-14**

Reviewer: Vasilis Pettas, University of Stuttgart

**Summary**

The manuscript presents the mid-fidelity wind farm simulator HAWC2Farm. It discusses the structure of the code and the basic principles of the modules (i.e. the individual turbine aeroelastic simulations, the collective wind fields and the superposition of the wake effects based on an extended version of the DWM) with a focus on wake-related aerodynamics. Moreover, a case study using measurements from the Lillgrund site is shown to validate the tool.

**General Comments**

The topic of the manuscript is relevant and nicely timed for the wind energy community as wind farm control and simulation is an important research topic currently under development. Mid-fidelity simulation tools that can capture in detail the aeroelastic response of each individual turbine both in terms of loads and power is a crucial step towards this goal.

The manuscript has a clear structure, is coherent, and in general well written while the topic fits the scope of the journal. I have some comments in the direction of the information passed to the reader and explaining a bit more some interesting points mentioned through the manuscript. My main concern is on the validation part which I think requires more quantitative results to serve its purpose. The current results can be considered as a general qualitative agreement with the measurements but they cannot be considered as a validation of the code.

Overall, I believe this is an interesting and relevant manuscript and should be published after a major revision addressing the comments mentioned below.

**Specific comments**

1. l.75 What does the term operating conditions mean in the context of HAWC2 outputs?
2. L 82 Is this recommendation referring to all dimensions of the box or only the rotor plane (YZ plane) here? Especially considering the turbine spacing in a farm this seems quite small.
3. L 86 This is not clear to me. Can you provide more information on how modifying the mean wind speed works? The Mann model (or the Veers for that matter) assumes stationarity for the wind speed. How can this work when the mean value is changed? Moreover, is there a limit in terms of spatial dimensions and duration beyond which these engineering models are not valid?
4. L 132 Can you explain to the uninformed reader what is LAPACK's xgstv routine?
5. L 142-143 This is a generally correct comment regarding implicit and explicit methods. Is it relevant for the levels of discretization used in wind farm simulation applications? I.e. since explicit methods are generally faster is there some recommendation on when they can be used?
6. L144-145 Is this correlated to the choice between implicit and explicit schemes? If so, can these be treated by applying some type of smoothening function to the obtained axial induction profiles? This could speed things up if it allows for an explicit scheme to be applied.
7. L146-147 What does random mean in this case? Are the profiles coming from Hawc2 or just assigning some random values in a range?

8. The study and discussion in section 2.3.1 regarding the choice of the solver are very interesting. Do you have any possible explanation for why this is happening? It seems both solvers behave almost the same until the 25 line but for larger ratios, the explicit solver seems to fail in every case. Can you also comment on the difference in computational time between the two methods?

9. L 165-172 What is the spatial filter shown with blue in figure 4? Is it the spatial averaging mentioned in l167? Clarify and explain briefly how it works.

10. L 232-233 It would be helpful for the community to list and discuss the measures taken to ensure numerical stability. The current statement is vague.

11. L 248-256 I recommend adding a table here stating the turbine names involved in each case and also the scenarios with their duration and scope. This will improve understanding for the reader in a compact manner.

12. L259-262 To understand the process better, add some more information on the lidar measurements (point-wise or rotor-effective speeds, sampling rate, location of measurement compared to the relevant turbine etc.) and the SCADA and load measurements (sampling rate, load sensors of interest etc.). Even if these are mentioned in other publications I think they should be also mentioned here briefly for completeness.

13. L268-269 Is it always the first turbine or the one upstream of the considered one?

14. L275-276 Can you explain why dividing with the standard deviation of the upstream turbine will tackle the problems mentioned here?

15. L289-290 As also discussed in the comments of section 2. Is a single 8-hour Mann box able to reproduce the conditions? I understand that the mean wind speed and probably turbulence and shear seem to be quite constant for the period, but the length and size of the box make me think that it might exceed the capabilities of turbulence boxes and their underlying assumptions. Also, how can short-term fluctuations like the ones between 17-18hr be captured when assuming a single stationary box? Would it make sense to cut the time series in smaller intervals and do multiple simulations or is there some other solution? I think this discussion is also of general interest and should be included here.

16. L291 Can you express TI also in terms of percentage? Same for l357. It will be helpful for readers not familiar with the Mann model formulation.

17. Sec 5.1.1 and 5.2.1 Provide some more information about the simulation that can ensure reproducibility and clarity. Turbulence box sizes and discretization in all directions, simulation time step, turbine model and controller (is it the one from the manufacturer or reversed engineered, etc.), communication interval between the wind farm controller and individual turbines, yaw actuator modelling, etc.

18. Sec 5.1.2 and sec 5.2.2 In the current state the results of all three scenarios are shown and discussed in figures 9, 10 and 13 which show the total time series of the total duration for power, nacelle direction, wind speed, blade root flap-wise moment and tower bottom side-side moment. The plots and discussion are interesting as a general overview of the capabilities of the simulation tool. Nevertheless, there is no quantitative comparison between the simulations and the measurements. This is done only through visual comparison of the long time series. This makes the argument of validation weak as the reader cannot understand at what level the response of each individual turbine is captured by the simulation tool. More quantitative results are required to consider this a validation exercise (especially for cases 1 and 2).

I recommend: either changing the scope of the paper avoiding referring to validation and keeping these results as a qualitative demonstration of the capabilities of the tool or adding more results that can be used to quantify the accuracy of the tool both in aerodynamic modelling and structural responses. For example, some metrics that can be used are: mean wind speed (or deficit), rotor

speed, power produced, DELs calculation, mean/std of the signal, etc. These can be considered for 10 min blocks throughout the whole duration of each scenario and compared directly with the measurements. This can show how well the outputs of the simulations agree with practical metrics that are used by the wind energy community. I understand that there might be high discrepancies due to model or measurement uncertainties but I strongly believe that validation requires also quantitative measures.

19. L350-351 Could be this attributed to a different shutdown control procedure? The loads seem quite different.

20. Sec 5 or Sec 6: I suggest adding some discussion on the computational time of the simulations. Information on wall time of simulation and scaling of time with increasing size and discretization. I think this is relevant information for the reader to understand the capabilities of the tool and the practical limits of the sizes and iterations that can be considered.

21. Sec6 As per the previous comments on validation, this section needs to be adjusted accordingly. Some strong statements regarding validation are not justified by the results shown.

**Minor corrections**

1. L 45 frozen synthetic turbulence box: rephrase
2. L 128 Equation
3. L 163 reference issue
4. L304 measured wind speed?

---

## Referee Comment (RC2)

**General comments**

This manuscript presents HAWC2Farm, an engineering tools that uses the Dynamic Wake Meandering framework to estimate the power outputs and loadings of a wind turbine in a farm, accounting for unsteady wake effects. The manuscript first presents the code and provide a description of the novelties introduced compared to other DWM implementations. An analysis of the performances is also proposed. In a second part, two scenarios from a real farm are reproduced and analysed.

This manuscript is consistent with the journal's editorial policy and tackles a major challenge of the wind energy field. I would thus advise to publish this paper, but some major modifications in the validation section must be made before, to improve the understandability and clarity of this work.

**Major comments**

- Paragraph 5.1.1 and 5.2.1: more informations on the turbulence intensity level are necessary to help the authors and readers to interpret the results.
  - $\alpha$ and $\epsilon$ at L291 are not defined. Also, does it means that there is 1% of turbulence intensity ?
  - If available, the TI measured by the LIDAR should be displayed and compared to the one of the simulation.
  - A bad TI value could explain discrepancies between measurements and simulations for power and wind speeds of a turbine in waked conditions.
- The relevancy of the first scenario before 18:00 is questionable. Indeed, as shown in Fig 7, there are large variations of wind speed, directions and yaw that are not captured by HAWC2. Therefore, it is difficult to analyse the results between 16:00 and 18:00. Please remove this part of the simulation, or give more details on that matter.
- The choice of normalisation makes it impossible to measure systematic error and it propagates errors of the first turbine to the other ones. The authors should use another normalisation or give a more in-depth justification of their choice.
  - Due to the normalisation, the reader cannot estimate the discrepancy described Line 315.
  - The choice of normalisation makes the reader believe that the wind speed is correct. However, there are actually discrepancies that lead the turbine not to work at the proper thrust and power coefficients.
  - The same issue comes back Line 225 where the authors describe some differences in wind speed that are seen in Figure 8 but not in Figure 9. The error induced by the discrepancies in inflow cannot be quantified.
    On that matter, Isn't it possible to apply a step in the inflow wind speed? There is about 1 m/s error after 22:00, which is not negligible.

**Minor comments**

- Please make a reference to Figure 1 in the text.
- In the last paragraph of section 2.3.2, it is mentioned that "50 random axial induction profiles" are used to trigger potential instabilities. Are these profiles totally random or are they built to be realistic? If yes, how?
- Can we have an estimation of the additional cost of the implicit solver? Is it negligible compared to the global cost of HAWC2Farm?

- Please give the colorbars on Figure 3.
- Line 175: does this statement means that the real cutoff frequency of DWM should be around U/(16D) instead of the classical value of U/(2D)? If yes, do you have any explanation of this difference?
- Please make a reference to Figure 5 in the text.
- Eq. 24 : given the results in Figures 9, 10 and 13, I believe the correct equation should be $\widehat{x_i} = \frac{x_i - \overline{x_0}}{\overline{x_0}}$ ? If not, the unit of the y-axis should be given to avoid confusion.
- Fig 9: please define Active Power.
- L 301: I think the discussed delay could also come from the choice of the advection velocity in the DWM. Values between $U_\infty$ and $0.8\, U_\infty$ have been debated in the literature, although it is not detailed here which is used in HAWC2.
- Paragraph 5.2.2: please cite the figure you are referring to.
- Figure 11: Please change the y-axis limits to the help read the picture. A change between 8 and 11 m/s is barely visible whereas it is consequent for the simulation, in particular near the rated.

**Technical corrections**

- To avoid double-parenthesis and inconsistencies when citing other works, I advise to use the \citep when out of text (e.g., *(Pederson et al. 2019)*, line 36) and \cite and inside the text (e.g., *Liew et al. (2022),* line 57).
- L46: I am not sure it is correct to use "time-stepping" as a verb.
- L118: missing the word *under* between "expressed" and "the following"
- L 163: There is an error of citation here
- L 294: I think you are referring to Figure 7, and not 8.
- L306: I would replace "in a fixed frame of reference" with "always at the same position" in order to avoid confusion with the moving and fixed frame of reference framework used in the DWM.
- Line 353: "Case and Case 2" should be replaced with "Case 1 and Case 2"

---

## Author Comment (AC1)

**Response to reviewers**

**Article title: Extending the Dynamic Wake Meandering Model in HAWC2Farm: Lillgrund Wind Farm Case Study and Validation**

We express our sincere appreciation to the two reviewers for their valuable and insightful feedback on our article. The authors have thoroughly reviewed and carefully considered the comments, and we firmly believe that their suggestions have significantly enhanced the quality of the document prior to publication.

To address the reviewer comments, we have made substantial revisions to all sections of the manuscript. These modifications were aimed at providing a clearer understanding of the methodology employed and facilitating a more comprehensive interpretation of the results. We are confident that the updated manuscript effectively addresses any uncertainties and greatly improves upon the previous version.

Please find below our detailed responses to your comments (highlighted in blue). Additionally, please find attached in the supplementary document a marked-up version showing all changes in the paper.

Yours Sincerely,

Jaime Liew, Tuhfe Göçmen, Alan Wai Hou Lio, Gunner Chr. Larsen

**Reviewer 1: Vasilis Pettas, University of Stuttgart**

**Summary**

The manuscript presents the mid-fidelity wind farm simulator HAWC2Farm. It discusses the structure of the code and the basic principles of the modules (i.e. the individual turbine aeroelastic simulations, the collective wind fields and the superposition of the wake effects based on an extended version of the DWM) with a focus on wake-related aerodynamics. Moreover, a case study using measurements from the Lillgrund site is shown to validate the tool.

**General Comments**

The topic of the manuscript is relevant and nicely timed for the wind energy community as wind farm control and simulation is an important research topic currently under development. Mid-fidelity simulation tools that can capture in detail the aeroelastic response of each individual turbine both in terms of loads and power is a crucial step towards this goal. The manuscript has a clear structure, is coherent, and in general well written while the topic fits the scope of the journal. I have some comments in the direction of the information passed to the reader and explaining a bit more some interesting points mentioned through the manuscript. My main concern is on the validation part which I think requires more quantitative results to serve its purpose. The current results can be considered as a general qualitative agreement with the measurements but they cannot be considered as a validation of the code. Overall, I believe this is an interesting and relevant manuscript and should be published after a major revision addressing the comments mentioned below.

**Specific comments**

1. l.75 What does the term operating conditions mean in the context of HAWC2 outputs?

In the context of HAWC2 outputs, "operating conditions" refers to the specific parameters that describe the wind turbine's state and performance. This includes power output (electrical power generated), rotor speed (blade rotation rate), and wind speed (velocity impacting the turbine). The document has been updated to better reflect this.

2. L 82 Is this recommendation referring to all dimensions of the box or only the rotor plane (YZ plane) here? Especially considering the turbine spacing in a farm this seems quite small.

Yes, the recommendation refers to all dimensions of the box, not just the rotor plane (YZ plane), as recommended in the paper (Liew J & Larsen, G. C. (2022, May). How does the quantity, resolution, and scaling of turbulence boxes affect aeroelastic simulation convergence?. In Journal of Physics: Conference Series (Vol. 2265, No. 3, p. 032049). IOP Publishing.). Although the specified dimensions may seem small, they are necessary to ensure converged structural load results.

3. L 86 This is not clear to me. Can you provide more information on how modifying the mean wind speed works? The Mann model (or the Veers for that matter) assumes stationarity for the wind speed. How can this work when the mean value is changed? Moreover, is there a limit in terms of spatial dimensions and duration beyond which these engineering models are not valid?

It is indeed possible to modify the wind speed as well by adjusting the propagation speed and the underlying ambient wind speed. It is important to note that modifying both wind direction and wind speed in this manner can potentially violate conservation laws. However, if these changes are executed gradually and carefully, they can still be approximately valid. The crucial thing is that the wind field shear is kept

constant, since the this is a basic assumption for the particular synthetic turbulence field realization. As for the limitations of engineering models like the Mann model or the Veers model, there may be spatial and temporal dimensions beyond which their validity decreases. The specific limits depend on various factors, such as the complexity of the wind flow and the accuracy requirements of the analysis.

4. L 132 Can you explain to the uninformed reader what is LAPACK's `xgstv` routine?

LAPACK's `xgtsv` function is tailored for efficient handling of systems of linear equations involving symmetric positive definite tridiagonal matrices, which aligns with the type of equation we are dealing with in this context. We have extended the section to better describe the use of the `xgtsv` routine for solving the wake profile equations.

5. L 142-143 This is a generally correct comment regarding implicit and explicit methods. Is it relevant for the levels of discretisation used in wind farm simulation applications? I.e. since explicit methods are generally faster is there some recommendation on when they can be used?

The manuscript has been revised to address the computational cost associated with both explicit and implicit solvers. While explicit methods are generally faster, it is important to consider the levels of discretisation used in wind farm simulation applications. In cases where the resolutions of axial induction over the blade are typical, the stable explicit discretisations may not cover the range of discretisations commonly encountered in aeroelastic simulations. Therefore, the use of explicit methods may not be recommended in such scenarios.

6. L144-145 Is this correlated to the choice between implicit and explicit schemes? If so, can these be treated by applying some type of smoothening function to the obtained axial induction profiles? This could speed things up if it allows for an explicit scheme to be applied.

This suggestion is indeed intriguing and reminiscent of the previous implementation in HAWC2, which utilized a 5-point stencil, as mentioned earlier in the manuscript. This approach involved performing an explicit solution with a smoothing step between each iteration and derivative operation. While it is possible to re-implement this approach, it is important to consider the potential trade-offs in terms of loss of detail and computational efficiency. Further investigation and analysis would be required to determine the feasibility and benefits of applying such a smoothing function to obtain axial induction profiles and potentially speed up the calculations using an explicit scheme.

7. L146-147 What does random mean in this case? Are the profiles coming from Hawc2 or just assigning some random values in a range?

In this case, the term "random" refers to the assignment of random values to the profiles. Specifically, the profiles are generated by assigning random numbers between -1 and 1. The intention behind using random profiles is to intentionally trigger numerical instabilities in the algorithm. This approach allows us to identify and analyse the stable regions within the system. Although the generated profiles may not represent physical scenarios, they serve a valuable purpose in assessing the stability of the algorithm under different conditions. The manuscript has been updated to describe this.

8. The study and discussion in section 2.3.1 regarding the choice of the solver are very interesting. Do you

have any possible explanation for why this is happening? It seems both solvers behave almost the same until the 25 line but for larger ratios, the explicit solver seems to fail in every case. Can you also comment on the difference in computational time between the two methods?

The stability criterion in this case shares similarities with other numerical schemes used to solve simplified versions of Navier-Stokes equations (e.g. Martínez-Tossas, Luis A., et al. "The aerodynamics of the curled wake: a simplified model in view of flow control." Wind Energy Science 4.1 (2019): 127-138). The criterion involves a factor of $\Delta x/\Delta r^2$. Due to the variable nature of U, V, and $\nu$ in the Dynamic Wake Meandering (DWM) definition of the wake profile, determining the exact stability criterion analytically becomes challenging. Therefore, a numerical approach was adopted to obtain a more representative stability map for the specific problem at hand.

The obtained results demonstrate a significant stability region for larger $\Delta x$ steps, which is often attributed to the implicit scheme's additional damping effects that contribute to stability. While the explicit scheme performed twice as fast for the same discretisation, the explicit scheme failed to accurately represent the wake with the required level of resolution for typical rotor simulations. Consequently, the computational cost became inconsequential since the explicit scheme could not fulfill the necessary task.

9. L 165-172 What is the spatial filter shown with blue in figure 4? Is it the spatial averaging mentioned in l167? Clarify and explain briefly how it works.

In this particular scenario, the spatial filter involves the uniform sampling of points throughout the area of the rotor disk at various longitudinal distances within the box. To ensure convergence, a sufficiently high resolution is employed, specifically using 2500 uniformly distributed points over the disk. The manuscript has been revised to provide the reader with a clearer definition of the spatial filter.

10. L 232-233 It would be helpful for the community to list and discuss the measures taken to ensure numerical stability. The current statement is vague.

Both this section and the section describing the stability of the algorithm (2.3.1) have been updated to better explain how stability is ensured. Namely, by choosing a suitable $\Delta x$ and $\Delta r$ pair.

11. L 248-256 I recommend adding a table here stating the turbine names involved in each case and also the scenarios with their duration and scope. This will improve understanding for the reader in a compact manner.

This is a helpful suggestion. The table outlines the availability of load or SCADA data, specifies the cases in which they are utilized, and indicates the turbines employed for normalization in the respective cases.

12. L259-262 To understand the process better, add some more information on the lidar measurements (point-wise or rotor-effective speeds, sampling rate, location of measurement compared to the relevant turbine etc.) and the SCADA and load measurements (sampling rate, load sensors of interest etc.). Even if these are mentioned in other publications I think they should be also mentioned here briefly for completeness.

Good suggestion. More details have been added to Section 4 regarding the measurements.

13. L268-269 Is it always the first turbine or the one upstream of the considered one?

*The normalising turbine was selected for each case to be the most upstream turbine. To clarify which turbine this is, the normalising turbines for each case are indicated in Table 2.*

14. L275-276 Can you explain why dividing with the standard deviation of the upstream turbine will tackle the problems mentioned here?

*The purpose of normalising by the standard deviation and the mean is to make the comparisons independent of two common miscalibrations in strain gauges: zero shift and span error. By normalising in this way, the load measurements become independent of the calibrated zero value and the modulus of elasticity of the strain sensors, enabling accurate analysis of load trends even in the presence of miscalibration. Section 5 has been updated to better describe this.*

15. L289-290 As also discussed in the comments of section 2. Is a single 8-hour Mann box able to reproduce the conditions? I understand that the mean wind speed and probably turbulence and shear seem to be quite constant for the period, but the length and size of the box make me think that it might exceed the capabilities of turbulence boxes and their underlying assumptions. Also, how can short-term fluctuations like the ones between 17-18hr be captured when assuming a single stationary box? Would it make sense to cut the time series in smaller intervals and do multiple simulations or is there some other solution? I think this discussion is also of general interest and should be included here.

*This is an insightful observation. Since a standard Mann box is stationary, the fundamental structure of the turbulence cannot be altered rapidly. However, certain aspects of the wind field can be modified to some extent. For instance, changes in wind direction can be simulated by rotating the Mann box and the overlapping wind fields, while variations in wind speed can be represented by adjusting the propagation speed of the turbulence box. Although these simplified representations may not capture the full complexity of the phenomena (which would occur across the entire wind farm), they still prove valuable to the wind energy community, particularly within aeroelastic simulation frameworks like HAWC2Farm, as they enable the approximate recreation of specific scenarios. In certain situations, it may be appropriate to split the simulation into multiple simulations. However, for the purpose of this demonstration paper, our objective was to showcase the flexibility of the simulation method in approximately simulating extended time periods, even when dealing with nonstationary events.*

16. L291 Can you express TI also in terms of percentage? Same for l357. It will be helpful for readers not familiar with the Mann model formulation.

*Turbulence intensity values have been provided to clarify the set up.*

17. Sec 5.1.1 and 5.2.1 Provide some more information about the simulation that can ensure reproducibility and clarity. Turbulence box sizes and discretization in all directions, simulation time step, turbine model and controller (is it the one from the manufacturer or reversed engineered, etc.), communication interval between the wind farm controller and individual turbines, yaw actuator modelling, etc.

*These details have now been included in both simulation setup sections.*

18. Sec 5.1.2 and sec 5.2.2 In the current state the results of all three scenarios are shown and discussed in figures 9, 10 and 13 which show the total time series of the total duration for power, nacelle direction, wind speed, blade root flap-wise moment and tower bottom side-side moment. The plots and discussion are interesting as a general overview of the capabilities of the simulation tool. Nevertheless, there is no quantitative comparison between the simulations and the measurements. This is done only through visual comparison of the long time series. This makes the argument of validation weak as the reader cannot understand at what level the response of each individual turbine is captured by the simulation tool. More quantitative results are required to consider this a validation exercise (especially for cases 1 and 2). I recommend: either changing the scope of the paper avoiding referring to validation and keeping these results as a qualitative demonstration of the capabilities of the tool or adding more results that can be used to quantify the accuracy of the tool both in aerodynamic modelling and structural responses. For example, some metrics that can be used are: mean wind speed (or deficit), rotorspeed, power produced, DELs calculation, mean/std of the signal, etc. These can be considered for 10 min blocks throughout the whole duration of each scenario and compared directly with the measurements. This can show how well the outputs of the simulations agree with practical metrics that are used by the wind energy community. I understand that there might be high discrepancies due to model or measurement uncertainties but I strongly believe that validation requires also quantitative measures.

The reviewer raises a good point regarding the scope of the work. The authors agree that the presented comparison leans towards qualitative results rather than quantitative. For this reason, as per your suggestion, we have decided to rescope the paper as a comparison with field measurements rather than a validation.

19. L350-351 Could be this attributed to a different shutdown control procedure? The loads seem quite different.

This could make sense if there were significant timing and duration differences between the measured and simulated shutdown sequence. The authors believe that the more likely explanation is in the calibration of the structural properties in the simulation as described in the conclusions.

20. Sec 5 or Sec 6: I suggest adding some discussion on the computational time of the simulations. Information on wall time of simulation and scaling of time with increasing size and discretization. I think this is relevant information for the reader to understand the capabilities of the tool and the practical limits of the sizes and iterations that can be considered.

This is a point which is brought up often, and would help clarify the computational resources required to run HAWC2Farm simulations. For this reason, Figure 6 has been added, which shows how a HAWC2Farm simulation duration compares to real-time as well as how it scales with number of turbines.

21. Sec 6 As per the previous comments on validation, this section needs to be adjusted accordingly. Some strong statements regarding validation are not justified by the results shown. Section 6 has been updated accordingly to better represent the rescoped manuscript.

**Minor corrections**

- L 45 frozen synthetic turbulence box: rephrase

- L 128 Equation

- L 163 reference issue

- L 304 measured wind speed?

Thanks for the suggestions. These corrections have been implemented in the new document.

**Reviewer 2**

**General comments**

This manuscript presents HAWC2Farm, an engineering tools that uses the Dynamic Wake Meandering framework to estimate the power outputs and loadings of a wind turbine in a farm, accounting for unsteady wake effects. The manuscript first presents the code and provide a description of the novelties introduced compared to other DWM implementations. An analysis of the performances is also proposed. In a second part, two scenarios from a real farm are reproduced and analysed.

This manuscript is consistent with the journal's editorial policy and tackles a major challenge of the wind energy field. I would thus advise to publish this paper, but some major modifications in the validation section must be made before, to improve the understandability and clarity of this work.

**Major comments**

Paragraph 5.1.1 and 5.2.1: more informations on the turbulence intensity level are necessary to help the authors and readers to interpret the results.

This is a helpful suggestion from the reviewer. Turbulence intensity information has been added to these two sections, as well as additional information regarding the simulation set up.

$\alpha$ and $\epsilon$ at L291 are not defined. Also, does it means that there is 1% of turbulence intensity ?

- If available, the TI measured by the LIDAR should be displayed and compared to the one of the simulation.
- A bad TI value could explain discrepancies between measurements and simulations for power and wind speeds of a turbine in waked conditions.

The reviewer is correct that the value $\alpha\epsilon^{2/3}$ does not correspond to that value in TI. The relationship between the two is complicated and depends on the size and discretisation of the turbulence box (see Liew, Jaime, and Gunner Chr Larsen. "How does the quantity, resolution, and scaling of turbulence boxes affect aeroelastic simulation convergence?." Journal of Physics: Conference Series.) TI values have now been added to allow readers who are not familiar with the Mann model to better understand the level of turbulence. In addition, $\alpha$ and $\epsilon$ have been defined. The authors also agree that a bad TI definition will cause discrepancies, which is why we have opted to tune the turbulence based on the value of $\alpha\epsilon^{2/3}$, also described in the aforementioned paper, to better match the load measurements. Nevertheless, this can still be the source of errors, which is further compounded by potential errors in the turbine structure calibration as mentioned in the conclusions.

The relevancy of the first scenario before 18:00 is questionable. Indeed, as shown in Fig 7, there are large variations of wind speed, directions and yaw that are not captured by HAWC2. Therefore, it is difficult to analyse the results between 16:00 and 18:00. Please remove this part of the simulation, or give more details on that matter.

The authors agree that the large variations in wind speed are not captured due to the way that this case is set up with constant wind speed. Nevertheless, the authors have chosen to include this part of the time series so that the entire duration of the yaw test is visible. While this particular part of the case does not match exactly, the behaviour between measurements and simulation show similar trends despite mismatches in the wind speed fluctuations.

The choice of normalisation makes it impossible to measure systematic error and it propagates errors of the first turbine to the other ones. The authors should use another normalisation or give a more in-depth justification of their choice.

- Due to the normalisation, the reader cannot estimate the discrepancy described Line 315.

- The choice of normalisation makes the reader believe that the wind speed is correct. However, there are actually discrepancies that lead the turbine not to work at the proper thrust and power coefficients.

- The same issue comes back Line 225 where the authors describe some differences in wind speed that are seen in Figure 8 but not in Figure 9. The error induced by the discrepancies in inflow cannot be quantified. On that matter, Isn't it possible to apply a step in the inflow wind speed? There is about 1 m/s error after 22:00, which is not negligible.

The first discrepancy that the reviewer is referring to is that of the turbines entering rated power output or not. This discrepancy remains noticeable despite the normalization process, as it becomes evident when the power output signal reaches its maximum threshold. Regarding the second aspect, the authors admit that there is a loss of information concerning the absolute wind speed values and potential discrepancies in the $C_T$ and $C_P$ curves. Nevertheless, it is reiterated that this normalization technique serves the purpose of comparing rows of turbines and assessing their relative behaviors, even in the presence of modelling and calibration biases. For this reason, we have chosen to keep the normalisation and to add more clarification to the start of the Results section to make it clear how these normalisations are defined and why they were chosen.

While we acknowledge the suggestion of introducing a step wind speed increase at 22:00 to potentially improve the matching of power output results, we would like to emphasize that the primary focus of this measurement campaign was on the step yaw sequence. Unfortunately, due to the deep location of the yawed turbine within the farm array, replicating it precisely was already challenging. Therefore, the authors believe that incorporating this additional step would not significantly enhance the comparison. Consequently, we have chosen not to redo the simulations but rather provide a thorough description of the discrepancy.

**Minor comments**

Please make a reference to Figure 1 in the text.

In the last paragraph of section 2.3.2, it is mentioned that "50 random axial induction profiles" are used to trigger potential instabilities. Are these profiles totally random or are they built to be realistic? If yes, how?

In this case, the term "random" refers to the assignment of random values to the profiles. Specifically, the profiles are generated by assigning random numbers between -1 and 1. The intention behind using random profiles is to intentionally trigger numerical instabilities in the algorithm. This approach allows us to identify and analyse the stable regions within the system. Although the generated profiles may not represent physical scenarios, they serve a valuable purpose in assessing the stability of the algorithm under different conditions. The manuscript has been updated to describe this.

Can we have an estimation of the additional cost of the implicit solver? Is it negligible compared to the global cost of HAWC2Farm?

The explicit scheme performed twice as fast as the implicit scheme for the same discretisation. However, the explicit scheme failed to accurately represent the wake with the required level of resolution for typical rotor simulations. Consequently, the computational cost became inconsequential since the explicit scheme could not fulfill the necessary task. Also, in comparison to the total HAWC2Farm simulation, the profile solver steps are indeed much smaller than the total computational time as you have mentioned. This section has been updated to make this point.

Please give the colorbars on Figure 3.

Colorbars have now been added.

Line 175: does this statement means that the real cutoff frequency of DWM should be around $U/(16D)$ instead of the classical value of $U/(2D)$? If yes, do you have any explanation of this difference?

The cut-off frequency $U/(16D)$ pertains to the temporal filter's cut-off frequency specifically associated with these box parameters. By utilising a temporal filter with this particular cut-off frequency, an equivalent frequency response can be achieved as that of a spatial filter with a $U/(2D)$ cut-off frequency. The required cut-off frequency to achieve this equivalence may vary depending on the box parameters. This section has been revised to provide a clearer explanation of this concept, along with a more detailed description of the process involved in determining the temporal cut-off frequency.

Please make a reference to Figure 5 in the text.

Figure 5 is now referenced in the code overview section (Section 3).

Eq. 24 : given the results in Figures 9, 10 and 13, I believe the correct equation should be $\hat{x}_l = \frac{x_i - \bar{x}_0}{\bar{x}_0}$? If not, the unit of the y-axis should be given to avoid confusion.

The normalisation equation employed is $\hat{x}_l = \frac{x_i - \bar{x}_0}{\sigma_0}$, specifically designed to address two prevalent causes of miscalibration in strain gauges: zero offset and span error. This normalization process enables the comparison of signals in a non-dimensional manner, even when unknown miscalibrations are present. The authors have expanded Section 5 to provide a comprehensive explanation of the rationale behind employing these normalisation techniques.

Fig 9: please define Active Power.

The term 'active power' has been relabeled to 'power output'. to better communicate that the channel refers to the power output of the turbine.

L 301: I think the discussed delay could also come from the choice of the advection velocity in the DWM. Values between $U_\infty$ and $0.8\,U_\infty$ have been debated in the literature, although it is not detailed here which is used in HAWC2.

The reviewer raises an interesting point. This has been considered in the implementation of the DWM model based on research on this topic (e.g. Andersen, Søren J., Jens N. Sørensen, and Robert F. Mikkelsen. "Turbulence and entrainment length scales in large wind farms"). In this particular scenario, the authors are confident that the wake propagation velocity is not responsible for the lag we see here. If the

wake propagation was slowed to, for example, $0.8U_\infty$, the delay seen at the start of Case 1 would be even more pronounced. The delay is also in the order of hours, which is slower than the propagation speed of the wakes. Instead, the authors attribute the discrepancy to the fluctuations in the wind speed at this time. In Figure 7 at around 17:00, the wind speed fluctuates at a generally higher value than the HAWC2Farm velocity, which likely causes B06 to experience a sudden rise in power output. This has been clarified in the results.

Paragraph 5.2.2: please cite the figure you are referring to.

Done.

Figure 11: Please change the y-axis limits to the help read the picture. A change between 8 and 11 m/s is barely visible whereas it is consequent for the simulation, in particular near the rated.

The y-axis limits have now been adjusted to more effectively portray the time series under consideration.

**Technical corrections**

To avoid double-parenthesis and inconsistencies when citing other works, I advise to use the \citep when out of text (e.g., (Pederson et al. 2019), line 36) and \cite and inside the text (e.g., Liew et al. (2022), line 57).

L46: I am not sure it is correct to use "time-stepping" as a verb.

Thank you for bringing up this point. You're right that 'time-stepping' is not typically used in this context. Instead, more commonly used terms such as 'time-marching' and 'stepwise' are used to convey simulations with discrete time steps. These alternatives have been used to replace 'time-stepping' in the manuscript.

L118: missing the word under between "expressed" and "the following"

L 163: There is an error of citation here

L 294: I think you are referring to Figure 7, and not 8.

L306: I would replace "in a fixed frame of reference" with "always at the same position" in order to avoid confusion with the moving and fixed frame of reference framework used in the DWM.

Line 353: "Case and Case 2" should be replaced with "Case 1 and Case 2"

All the above corrections have now been rectified.

[revised manuscript text omitted]

---

## Author Response (AR2)

**Response to reviewers**

**Article title: Extending the Dynamic Wake Meandering Model in HAWC2Farm: Lillgrund Wind Farm Case Study and Validation**

We express our sincere gratitude to the two reviewers for their valuable and insightful feedback on our article. Please find our responses to the remaining minor comments (highlighted in blue). Additionally, please find attached in the supplementary document a marked-up version highlighting minor changes made in the manuscript.

Yours Sincerely,

Jaime Liew, Tuhfe Göçmen, Alan Wai Hou Lio, Gunner Chr. Larsen

**Reviewer 2**

**Comments**

Most of the remarks given in the first round of review were properly addressed. I have only two remaining minor remarks:

I am still perturbed by the choice of normalisation by the authors. I understand their justification, which makes sense, but I think it means that the present validation should be accompanied (if it doesn't exist already) with another validation that focuses on systematic error. Maybe an additional word or two would be needed here.

While we have provided a rationale for our chosen normalization method, we acknowledge the importance of further validation that specifically addresses systematic error. Unfortunately, due to limitations in the available measurement data and the control campaigns, a more comprehensive quantitative comparison was not feasible, as discussed in the manuscript's conclusions section.

We agree with the reviewer that additional comparisons would be valuable to complement the presented findings. However, we believe that the existing comparison offers a valuable demonstration of the dynamic modeling of wind farms, showcasing the strengths and weaknesses of the presented methodology. We have placed increased emphasis on the limitations associated with the employed normalisations in the revised manuscript.

I also think that the choice of the threshold at -3dB that leads to a cutoff frequency of u/16D is arbitrary, and maybe choosing a threshold at -5dB for instance would have led to a different result. If possible, maybe the authors should discuss a bit more about that. Even though it is a minor point that is already discussed by the authors, I am worried that a "quick reader" can draw wrong conclusions on that part.

We understand your concern, and we would like to clarify that the -3dB threshold is a commonly accepted standard in signal processing. It represents the point at which the power or amplitude is reduced by approximately half, which is widely used to define the cutoff frequency in filter design. That is,

$$\log_{10} \left( \frac{1}{\sqrt{2}} \right) = 3.0103 dB \approx 3 dB \tag{1}$$

The wording of this section of the manuscript has been modified to indicate this for readers who are not familiar with the signal processing convention.

[revised manuscript text omitted]